# Vestigial singlet pairing in a fluctuating magnetic triplet superconductor and its implications for graphene superlattices

Prathyush P. Poduval[1,2] ✉ & Mathias S. Scheurer [3,4] ✉

Stacking and twisting graphene layers allows to create and control a two-dimensional electron liquid with strong correlations. Experiments indicate that these systems exhibit strong tendencies towards both magnetism and triplet superconductivity. Motivated by this phenomenology, we study a 2D model of fluctuating triplet pairing and spin magnetism. Individually, their respective order parameters, $d$ and $N$, cannot order at finite temperature. Nonetheless, the model exhibits a variety of vestigial phases, including charge-$4e$ superconductivity and broken time-reversal symmetry. Our main focus is on a phase characterized by finite $d \cdot N$, which has the same symmetries as the BCS state, a Meissner effect, and metastable supercurrents, yet rather different spectral properties: most notably, the suppression of the electronic density of states at the Fermi level can resemble that of either a fully gapped or nodal superconductor, depending on parameters. This provides a possible explanation for recent tunneling experiments in the superconducting phase of graphene moiré systems.

Strongly correlated systems often exhibit complex phase diagrams with multiple phases, characterized by long-range or quasi-long-range order (QLRO) of different order parameters. Aside from phase competition as a possible origin, a rich set of phases might also be understood as different manifestations of an underlying primary order —a concept often referred to as "intertwined orders"[1]. For instance, thermal or quantum fluctuations can disorder a primary order parameter, while higher-order composite order parameters can still survive. An example of such a "vestigial phase"[2,3] is the charge-$4e$ superconducting state that emerges when a charge-$2e$ pair density wave order parameter, $\Delta_Q$, itself vanishes, yet $\Delta_Q \Delta_{-Q}$ does not[4]; this and other forms of charge-$4e$ superconductivity have attracted a lot of attention[5–18], in particular, as a result of recent experiments[19,20].

Another exciting recent development is the emergence of twisted graphene moiré superlattices as versatile playgrounds for strongly correlated physics[21,22]. These systems display a variety of different phases such as nematic[23–25] and density-wave order[26–28], different forms of magnetism[29–33], and, possibly unconventional[34,35]

superconductivity[36]; magnetism and superconductivity appear in the same density range[34,35,37–41] and recent experiments[33,42] demonstrate that they can coexist microscopically. Motivated by these observations, we here study the case of two primary order parameters: a fully gapped spin–triplet superconductor ($d$), which could explain[43] the subgap states at strong tunneling in recent experiments[35], and, in line with the conclusions of[41,44], magnetic order ($N$) with antiparallel spins in the two valleys. At finite temperature, $T > 0$, it must hold $\langle d \rangle = \langle N \rangle = 0$ in two dimensions (2D). However, there are several different vestigial phases characterized by the composite order parameters $\phi_{dd} = d \cdot d$, $\phi_{dN} = d \cdot N$, and $\phi_{ddN} = i(d^\dagger \times d) \cdot N$. These include not only a charge-$4e$ superconductor[45,46], see Fig. 1a, but also a charge-$2e$ state, which has the same symmetries as and is, hence, adiabatically connected to the BCS state. However, it should primarily be thought of as a condensate of three electrons and a hole, see Fig. 1b, or, more formally, QLRO of $\phi_{dN}$. We develop a theory for this state and study its spectral properties at finite $T$, which are rather different from those of the BCS state. Depending on $T$ and $\phi_{dN}$, we obtain a low-energy suppression of the

[1]Donald Bren School of Information and Computer Sciences, University of California, Irvine, CA 92697, USA. [2]Condensed Matter Theory Center, Department of Physics, University of Maryland, College Park, MD 20742, USA. [3]Institute for Theoretical Physics III, University of Stuttgart, 70550 Stuttgart, Germany. [4]Institute for Theoretical Physics, University of Innsbruck, Innsbruck A-6020, Austria. ✉e-mail: ppoduval@uci.edu; mathias.scheurer@itp3.uni-stuttgart.de

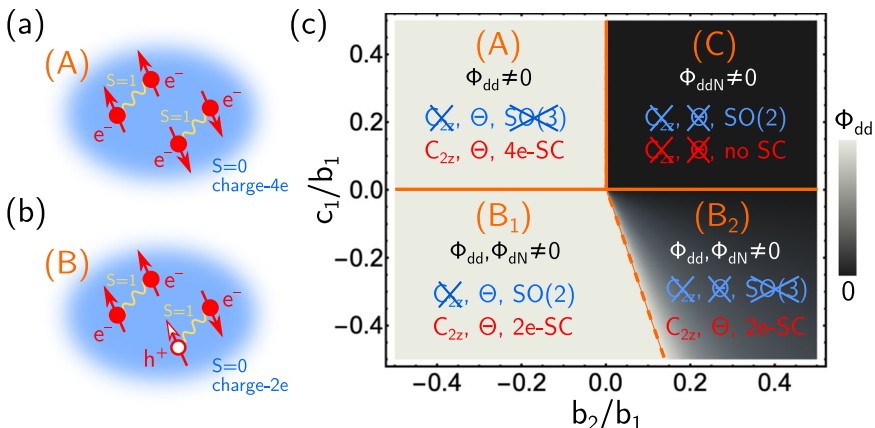

**Fig. 1 | Possible phases.** We illustrate schematically the two finite-$T$ vestigial superconducting phases of our model, **a** a charge-$4e$ state where two pairs of electrons each in a triplet state condense, and **b** where three electrons and a hole pair; in both cases, the objects that condense are spin singlets in agreement with the Mermin–Wagner theorem. **c** Mean-field phase diagram for $r_d = r_N$, $b_3 = b_1$, $c_2 = 0$, where we indicate the symmetries at $T = 0$ (blue), those of the resulting vestigial phases at $T > 0$ (red), and which composite order parameters are finite. Solid (dashed) orange lines are phase transitions at $T = 0$ and $T > 0$ (become a crossover at $T > 0$).

density of states (DOS) similar to a fully gaped or nodal state. This could provide an alternative explanation[43,44,47,48] to the tunneling data of [34,35], which does not require any momentum dependence in the superconducting order parameter.

## Results

### Model

We consider a 2D model, which is similar in spirit to the celebrated spin fermion model[49], and describes both triplet superconductivity and spin magnetism, with three-component order parameter fields $\boldsymbol{d}$ (complex) and $\boldsymbol{N}$ (real), respectively. Denoting the electronic field operators of spin $s = \uparrow, \downarrow$ (Pauli matrices $\boldsymbol{s}$) and in valley $\tau = \pm$ (Pauli matrices $\boldsymbol{\tau}$) by $c_{k,s,\tau}$, where $k = (i\omega_n, \boldsymbol{k})$ comprises Matsubara frequencies and 2D momentum, they couple as

$$S_c = \lambda \int_{k,q} \left[ c_{k-q}^\dagger \boldsymbol{s} \boldsymbol{N}_q \tau_z c_k + (c_{k-q}^\dagger \boldsymbol{s} \boldsymbol{d}_q i s_y \tau_y c_{-k}^\dagger + \text{H.c.}) \right].$$

Note that $\boldsymbol{N}$ couples anti-ferromagnetically in the two valleys; while the ferromagnetic case can be studied similarly, we focus on antiferromagnetism not only for concreteness here but also because recent microwave experiments[41] and multiple other experiments[44] favor this scenario. The bare dynamics of $\boldsymbol{d}$ and $\boldsymbol{N}$ is governed by

$$S_\chi = \int_q \left[ \chi_N^{-1}(q) \boldsymbol{N}_q \boldsymbol{N}_{-q} + \chi_d^{-1}(q) \boldsymbol{d}_q^\dagger \boldsymbol{d}_q \right].$$

We take the susceptibilitites to be $\chi_\mu(q) = \bar{\chi}_\mu / (r_\mu + \Omega_n^2 + v_\mu^2 \boldsymbol{q}^2)$, $\mu = N, d$, where $q = (i\Omega_n, \boldsymbol{q})$ and $\Omega_n$ are bosonic Matsubara frequencies. Up to quartic order, the local bosonic interactions allowed by the symmetries listed in Table 1 can be written as $S_V = \int_x V(\boldsymbol{d}(x), \boldsymbol{N}(x))$ with

$$V = b_1 (\boldsymbol{d}^\dagger \boldsymbol{d})^2 + b_2 |\boldsymbol{d} \boldsymbol{d}|^2 + b_3 \boldsymbol{N}^4 + c_1 |\boldsymbol{d} \boldsymbol{N}|^2 + c_2 (\boldsymbol{d}^\dagger \boldsymbol{d}) \boldsymbol{N}^2. \tag{1}$$

Finally, the bare electronic action is given by $S_e = \int_k c_{k,\tau,s}^\dagger (-i\omega_n + \epsilon_{\tau \cdot \boldsymbol{k}}) c_{k,\tau,s}$.

### Zero-temperature phases

To gain an overview of the different possible phases at zero temperature, we start with a simple mean-field analysis, which proceeds by minimizing the potential term $S_V$ for different values of $b_{1,2,3}$ and $c_{1,2}$. Assuming that both $\langle \boldsymbol{d} \rangle$ and $\langle \boldsymbol{N} \rangle$ are non-zero and homogeneous, we obtain the four distinct zero-temperature phases labeled $(A)$, $(B_{1,2})$, and

**Table 1 | Relevant symmetries $g$ and their action on the field operators**

| $g$ | $c_k$ | $\boldsymbol{N}$ | $\boldsymbol{d}$ | $\phi_{dd}$ | $\phi_{dN}$ | $\phi_{ddN}$ |
|---|---|---|---|---|---|---|
| $U(1)$ | $e^{i\varphi} c_k$ | $\boldsymbol{N}$ | $e^{-2i\varphi} \boldsymbol{d}$ | $e^{-4i\varphi} \phi_{dd}$ | $e^{-2i\varphi} \phi_{dN}$ | $\phi_{ddN}$ |
| $SO(3)$ | $e^{i\varphi \cdot \boldsymbol{s}} c_k$ | $R_\varphi \boldsymbol{N}$ | $R_\varphi \boldsymbol{d}$ | $\phi_{dd}$ | $\phi_{dN}$ | $\phi_{ddN}$ |
| $C_{2z}$ | $\tau_x c_{-k}$ | $-\boldsymbol{N}$ | $-\boldsymbol{d}$ | $\phi_{dd}$ | $\phi_{dN}$ | $-\phi_{ddN}$ |
| $\Theta$ | $is_y \tau_x c_{-k}$ | $\boldsymbol{N}$ | $-\boldsymbol{d}^\dagger$ | $\phi_{dd}^*$ | $-\phi_{dN}^*$ | $-\phi_{ddN}$ |

Here $R_\varphi$ is the orthogonal matrix obeying $e^{-i\varphi \cdot \boldsymbol{s}} \boldsymbol{s} e^{i\varphi \cdot \boldsymbol{s}} = R(\varphi) \boldsymbol{s}$. All symmetries are linear except for $\Theta$, which is anti-linear.

$(C)$ in Fig. 1c, with respective symmetries indicated in blue (cf. Table 1). We note that there is no relation of the labeling of the phases we use here to the nomenclature in superfluid ³He. Using $\hat{\boldsymbol{e}}_{1,2,3} \in \mathbb{R}^3$ to denote orthogonal unit vectors, we have $\boldsymbol{N} = N_0 \hat{\boldsymbol{e}}_1$ and $\boldsymbol{d} = d_0 e^{i\alpha} \hat{\boldsymbol{e}}_2$ in phase (A), which breaks $SO(3)$ completely, while $\Theta$ is preserved (in any gauge-invariant observable); as for any phase with $\langle \boldsymbol{N} \rangle \neq 0$, $C_{2z}$ is broken. In phase $(B_1)$, $\boldsymbol{N}$ and $\boldsymbol{d}$ are aligned; we, thus, obtain a residual spin-rotation symmetry $SO(2)$ along that direction and $\Theta$ is preserved too. Beyond a critical value of $b_2$, an additional component with relative phase $\pi/2$ emerges in $\boldsymbol{d}$, defining phase $(B_2)$ where $\boldsymbol{N} = N_0 \hat{\boldsymbol{e}}_1$ and $\boldsymbol{d} = d_0 e^{i\alpha} (\hat{\boldsymbol{e}}_1 + i\eta \hat{\boldsymbol{e}}_2)$, with $0 < \eta < 1$; this is a distinct phase as $\eta \neq 0$ breaks both the residual $SO(2)$ spin symmetry and $\Theta$. Finally, phase (C) is characterized by $\boldsymbol{N} = N_0 \hat{\boldsymbol{e}}_1$ and $\boldsymbol{d} = d_0 e^{i\alpha} (\hat{\boldsymbol{e}}_2 + i\hat{\boldsymbol{e}}_3)$. Consequently, $\Theta$ is also broken but a residual $SO(2)$ spin-symmetry remains. We finally point out that the location of the different phases in Fig. 1(c) is straightforward to understand intuitively. Positive (negative) $c_1$ disfavors (favors) alignment of $\boldsymbol{d}$ along $\boldsymbol{N}$, which is why these two vectors are perpendicular in phase (A) and (C) [are (partially) aligned in $(B_{1,2})$]. What is more, negative (positive) $b_2$ prefers a unitary (non-unitary) triplet component to maximize (minimize) $|\boldsymbol{d} \boldsymbol{d}|$ at a fixed length of $\boldsymbol{d}$.

### Vestigial phases at finite $T$

Importantly, $\langle \boldsymbol{d} \rangle, \langle \boldsymbol{N} \rangle \neq 0$ is not possible for finite $T$, and thus, our discussion of symmetries and phases above is only valid for $T = 0$ in 2D. Nonetheless, the $T = 0$ results will help us understand the possible vestigial phases at finite temperatures, $T > 0$, in our model, which will be the focus of the remainder of this paper. To study $T > 0$, where $SO(3)$ spin-rotation symmetry is preserved and $\langle \boldsymbol{d} \rangle = \langle \boldsymbol{N} \rangle = 0$, it is convenient to define the following composite order parameters $\phi_{dd} = \boldsymbol{d} \cdot \boldsymbol{d}$, $\phi_{dN} = \boldsymbol{d} \cdot \boldsymbol{N}$, and $\phi_{ddN} = i(\boldsymbol{d}^\dagger \times \boldsymbol{d}) \cdot \boldsymbol{N}$, with symmetry properties listed in Table 1. These are constructed as the lowest-order, local combinations

of the bosonic fields $\boldsymbol{d}$ and $\boldsymbol{N}$ that are invariant under SO(3) spin-rotations while transforming non-trivially under U(1)-gauge or the point symmetries. By virtue of being SO(3) invariant, they can exhibit long-range (in case of $\phi_{ddN}$) or QLRO (in case of $\phi_{dd}$, $\phi_{dN}$) at finite $T$. We indicate this in Fig. 1c for the different phases. This immediately tells us that, in spite of $\langle \boldsymbol{d} \rangle = 0$, phase (A) transitions for finite $T$ into a state where $\phi_{dd}$ has QLRO and, thus, constitutes a charge-4$e$ superconductor (as $\phi_{dN} = 0$), which does not break $C_{2z}$ or $\Theta$ (as $\phi_{ddN} = 0$); intuitively, one can think of this state as a condensate of four electrons forming a spin-singlet out of two triplets, see Fig. 1a. At finite $T$, $(B_1)$ and $(B_2)$ will both preserve all normal-state symmetries and become the same phase, which we denote by $(B)$ in the following. It is characterized by QLRO not only in $\phi_{dd}$ but also in $\phi_{dN}$; as the latter has charge 2$e$, it is a charge-2$e$ superconductor and adiabatically connected to the conventional BCS state. Nonetheless, in our current description, this state should rather be thought of as the condensation of three electrons and a hole, see Fig. 1b, consisting of a pair of electrons in a triplet state forming a singlet with a spin-1 particle-hole excitation. In fact, we will see below that it exhibits spectral properties rather different from those of the BCS state at finite $T$. Finally, while phase $(C)$ does not exhibit any vestigial pairing at $T > 0$, it will have long-range order in $\phi_{ddN}$ and, as such, continues to break both $C_{2z}$ and $\Theta$.

## Theory for phase (B)

As $c_1 < 0$ is found when the coefficients in $V$ are computed from high-energy electronic degrees of freedom within the mean-field theory (see Methods), we focus on phase (B). To obtain an efficient description of this phase that properly captures the preserved SO(3) symmetry at finite temperature, we first decouple the four terms in $V$ using four Hubbard–Stratonovich fields, $\psi_d$ for $\boldsymbol{d}^\dagger \boldsymbol{d}$, $\psi_N$ for $\boldsymbol{N}^2$, $\phi_d$ for $\boldsymbol{d} \cdot \boldsymbol{d}$, and $\phi_{dN}$ for $\boldsymbol{d} \cdot \boldsymbol{N}$. We treat them on the saddle-point level, which becomes exact in the limit where the number of components of $\boldsymbol{d}$ and $\boldsymbol{N}$ is taken to be infinitely large[50]. This procedure does not violate Mermin–Wagner's theorem (as opposed to taking the limit of infinitely many fermion flavors). The saddle point values of $\psi_d$ and $\psi_N$ will in general, be non-zero, which we absorb into a redefinition of $r_{d,N}$. Then, the effective action for phase (B) becomes $\mathcal{S}_B = \mathcal{S}_\chi + \mathcal{S}_e + \mathcal{S}_c + \mathcal{S}_\phi$ where the key new component reads as

$$\mathcal{S}_\phi = \int_q \left[ \phi^0_{dN} \boldsymbol{d}_q \cdot \boldsymbol{N}_{-q} + \phi^0_{dd} \boldsymbol{d}_q \cdot \boldsymbol{d}_{-q} + \text{H.c.} \right]. \quad (2)$$

While generically, both saddle point values $\phi^0_{dN}$ and $\phi^0_{dd}$ are expected to be non-zero simultaneously in phase (B), we take $\phi^0_{dd} \to 0$ and $\phi^0_{dN} \equiv \phi_0 \neq 0$ for the following explicit calculations to model the QLRO of phase (B). Setting $\phi^0_{dd} = 0$ does not change any symmetries of the phase, allows for a more compact discussion of the results, and can formally be seen as the large $b_2$ limit of the theory where $\phi^0_{dd}$ is suppressed [cf. Fig. 1c]. More generally than its derivation via Hubbard–Stratonovich transformations, $\mathcal{S}_B$ can also be thought of as the simplest field theory capturing the key aspects of phase (B) in Fig. 1c at finite $T$. Crucially, this is still an interacting theory, and integrating out the bosons yields the effective fermionic description $\mathcal{S}'_B = \mathcal{S}_e + \mathcal{S}_1 + \mathcal{S}_2$; here $\mathcal{S}_{1,2}$ are the following four-fermion interactions

$$\mathcal{S}_1 = -\int_q \frac{\lambda^2}{M_q} \left( \frac{\chi_d^{-1}}{4} \boldsymbol{S}_q \cdot \boldsymbol{S}_{-q} + \chi_N^{-1} \boldsymbol{D}_q \cdot \boldsymbol{D}_q^\dagger \right), \quad (3a)$$

$$\mathcal{S}_2 = -\frac{1}{2} \int_q \frac{\lambda^2}{M_q} \left( \phi_0 \, \boldsymbol{S}_q \cdot \boldsymbol{D}_q^\dagger + \phi_0^* \, \boldsymbol{D}_q \cdot \boldsymbol{S}_{-q} \right), \quad (3b)$$

with $M_q = \chi_d^{-1} \chi_N^{-1} - |\phi_0|^2$ and where we introduced the fermionic bilinears $\boldsymbol{S}_q = \int_k c_{k+q}^\dagger \boldsymbol{s} \tau_z c_k$ and $\boldsymbol{D}_q = \int_k c_{k+q}^\dagger i s_y \tau_y c_{-k}^\dagger$ to write Eq. (3) in

more compact form. The two terms in $\mathcal{S}_1$ describe spin and superconducting triplet fluctuations, respectively, while $\mathcal{S}_2$ captures the Higgs mechanism underlying phase (B)'s superconducting phenomenology, to be discussed below. Note that $M_q \to \chi_d^{-1} \chi_N^{-1}$ for $\phi_0 \to 0$ such that the first and second terms in Eq. (3a) are proportional to $\chi_N$ and $\chi_d$, respectively, as expected.

## Minimal mean-field theory

Before analyzing $\mathcal{S}'$ in a more systematic way below, we first study a simple, effective Hamiltonian associated with setting $q = 0$ in the particle-number-non-conserving interaction term in Eq. (3b); we treat it within self-consistent Hartree–Fock, only allowing for spin-rotation invariant operators to condense (see Supplementary Appendix A). This will help us to understand the behavior of the electronic spectrum in phase (B) more intuitively in a minimal setting. The effective (static $i\Omega = 0$) interaction is given by

$$\int_{k_1, k_2} \left[ \frac{g}{6} \left( \Psi_{k_1}^\dagger \boldsymbol{s} \gamma_z \Psi_{k_1} \right) \cdot \left( \Psi_{k_2}^\dagger \boldsymbol{s} i \gamma_- \Psi_{k_2} \right) + \text{H.c.} \right], \quad (4)$$

where we introduced the Nambu-spinor $\Psi_k = (c_{k,+}, i s_y c_{-k,-}^\dagger)^T$, with associated Pauli matrices $\gamma_i$ in Nambu space and the complex-valued coupling constant

$$g = -\frac{6 \lambda^2 r_N \phi_0}{v_N^2 (r_N r_d - |\phi_0|^2)}. \quad (5)$$

The free Hamiltonian reads as $H_0 = \int_k \Psi_k^\dagger \epsilon_k \gamma_z \Psi_k$. Performing a mean-field decomposition of the interaction in Eq. (4) that retains spin-rotation invariance in line with the Mermin–Wagner theorem, we arrive at

$$H_{\text{MF}} = \int_k \Psi_k^\dagger \left[ \epsilon_k \gamma_z - \frac{1}{2} (g \, \gamma_y C_k \gamma_z + \text{H.c.}) \right] \Psi_k \quad (6)$$

$$\equiv \int_k \Psi_k^\dagger \left[ \tilde{\epsilon}_k \gamma_z + \text{Im} \, \tilde{\Delta}_k \gamma_y + \text{Re} \, \tilde{\Delta}_k \gamma_x \right] \Psi_k, \quad (7)$$

where $C_k = -\langle \Psi_k \Psi_k^\dagger \rangle$, and $\tilde{\epsilon}_k, \tilde{\Delta}_k$ are the self-consistent band structure and superconducting singlet order parameter. The resulting self-consistency equations become

$$\tilde{\epsilon}_k = \epsilon_k + \tilde{\Delta}_k \beta_k^* \quad (8a)$$

$$\tilde{\Delta}_k = \tilde{\epsilon}_k \beta_k, \quad \beta_k = g \frac{\tanh\left(\frac{E_k}{2T}\right)}{2 E_k}, \quad (8b)$$

where $E_k = \sqrt{\tilde{\epsilon}_k^2 + |\tilde{\Delta}_k|^2}$. The solutions of these self-consistency equations contain a strong temperature dependence. When $T > |g|/4$, the coupling $|\beta_k|$ is smaller than 1 for all $\boldsymbol{k}$. Thus, the self-consistency equations can be rearranged as

$$\tilde{\epsilon}_k = \frac{1}{1 - |\beta_k|^2} \epsilon_k, \quad \tilde{\Delta}_k = \frac{\beta_k}{1 - |\beta_k|^2} \epsilon_k, \quad (9)$$

which gives us $E_k = \frac{\sqrt{1 + |\beta_k|^2}}{1 - |\beta_k|^2} \epsilon_k$. In this regime, the self-consistent solution simply renormalizes the Fermi velocity, with states near the Fermi surface pushed away from it. This is in stark contrast to the behavior in the usual, BCS-like, mean-field theory of superconductivity. The difference is tied to the fact that, to linear order, $\tilde{\Delta}_k$ only appears on one side of the self-consistency equations in Eq. (8), which leads to $\tilde{\Delta}_k \propto \epsilon_k$ in Eq. (9); intuitively, this comes from the extra factor

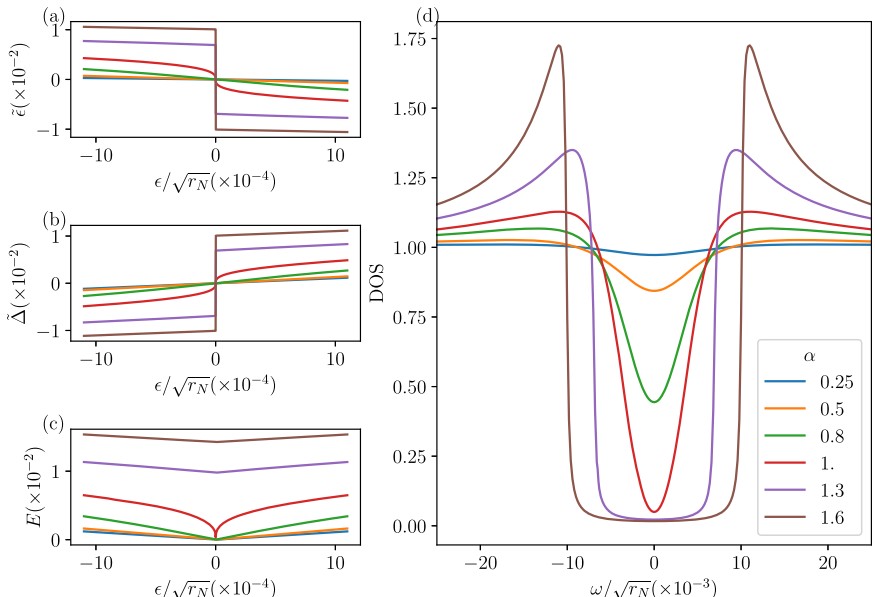

**Fig. 2 | Self-consistent Hartree–Fock. a–c** The renormalized band parameters $\tilde{\epsilon}, \tilde{\Delta}, E$ as a function of the free energy $\epsilon$ within self-consistent Hartree–Fock. For $\alpha < 1$, $\tilde{\epsilon}$ and $\tilde{\Delta}$ vary linearly with $\epsilon$ near $\epsilon = 0$. The corresponding slope diverges as $\alpha \to 1$, and becomes non-analytic at $\alpha = 1$, developing a discontinuity in the band parameters at $\epsilon = 0$ for $\alpha > 1$, resulting in a hard gap. **(d)** The corresponding density of states within self-consistent Hartree–Fock, very similar behavior is found with an explicit hard gap for $\alpha > 1$.

of $c^\dagger c$ in the interaction $\phi_0 c^\dagger c^\dagger c + \text{H.c.} = c^\dagger c (\phi_0 c^\dagger c^\dagger + \text{H.c.})$ that we decouple. More formally, it is clear that this cannot arise in BCS-like mean-field theory of superconductivity since the superconducting order parameter is the only complex number in the theory that transforms non-trivially under U(1) gauge transformations, $c \to e^{i\varphi} c$, requiring it to appear on both sides of the self-consistency equations. In our case, this is different since also $g \to e^{-2i\varphi} g$ and, hence, $\beta_{\boldsymbol{k}} \to e^{-2i\varphi} \beta_{\boldsymbol{k}}$, allowing the behavior $\tilde{\Delta}_{\boldsymbol{k}} \propto \beta_{\boldsymbol{k}} \epsilon_{\boldsymbol{k}}$ in Eq. (9).

Since $\beta_{\boldsymbol{k}}$ also depends on $E_{\boldsymbol{k}}$, $E_{\boldsymbol{k}} = \frac{\sqrt{1+|\beta_{\boldsymbol{k}}|^2}}{1-|\beta_{\boldsymbol{k}}|^2} \epsilon_{\boldsymbol{k}}$ should be thought of as a self-consistency equation to be solved for $\beta_{\boldsymbol{k}}$ or $E_{\boldsymbol{k}}$. Nonetheless, it allows to readily derive asymptotic relations. In the limit $\epsilon_{\boldsymbol{k}} \to 0$, we have $E_{\boldsymbol{k}} \to 0$ and $\beta_{\boldsymbol{k}} \to \frac{|g|}{4T} =: \alpha < 1$, ensuring the expressions in Eq. (9) are well behaved for $\epsilon_{\boldsymbol{k}} \to 0$. Near $\epsilon_{\boldsymbol{k}} = 0$ and for large $T \gg |g|$, i.e., $\alpha \ll 1$, the renormalized spectrum is given by $E_{\boldsymbol{k}} \simeq \sqrt{1 + \frac{3|g|^2}{16T^2}} \epsilon_{\boldsymbol{k}}$; the associated temperature-dependent reduction of the electronic mass reduces the DOS and leads to an expression of the same form that we will derive diagrammatically in Eq. (11) below.

When $T/|g| \to 1/4^+$ (i.e., $\alpha \to 1^-$), we have $|\beta_{\boldsymbol{k}}|^2 \to 1$ as we approach the Fermi surface $\epsilon_{\boldsymbol{k}} \to 0$, and the expressions in Eq. (9) are not valid anymore because $1 - |\beta_{\boldsymbol{k}}|^2$ is not invertible on the Fermi surface. At this temperature and for lower temperatures, the self-consistent solutions open up a gap in $E_{\boldsymbol{k}}$ when $\epsilon_{\boldsymbol{k}} = 0$, which we find by solving the equation $|\beta_{\boldsymbol{k}}|^2 = 1$, giving us $E_{\boldsymbol{k}}|_{\epsilon_{\boldsymbol{k}}=0} \sim \sqrt{12}T\sqrt{1 - \frac{4T}{|g|}}$ as $T$ approaches $|g|/4$ from below.

In Fig. 2, we illustrate the self-consistent mean-field solutions. Figure 2a–c shows the renormalized spectrum $\tilde{\epsilon}_{\boldsymbol{k}}, \tilde{\Delta}_{\boldsymbol{k}}$ and $E_{\boldsymbol{k}}$ respectively, as a function of $\epsilon_{\boldsymbol{k}}$ for various values of $\alpha$. As $\alpha$ approaches 1 from below, the slope of $\tilde{\epsilon}(\epsilon)$ approaches $\infty$ at $\epsilon = 0$, and $\tilde{\epsilon}(\epsilon)$, becomes non-analytic at $\alpha = 1$ At $\alpha > 1$, the non-analyticity at $\epsilon = 0$ turns into a discontinuity, with the self-consistent solutions developing a finite gap; we have checked numerically that including finite momentum transfer in the mean-field equations regularizes this non-analytic behavior. Figure 2d shows the resulting DOS in our minimal mean-field model: for small $\alpha$, one finds only a partial suppression of the low-energy spectral weight, in line with our expansion in small $\alpha$ discussed above

and Eq. (11) below; including higher-order corrections leads to a hard gap for $\alpha \geq 1$. At $T = 0$, we find $|\tilde{\Delta}| = |\tilde{\epsilon}| = 2^{-3/2}|g|$ for $\epsilon = 0$, which corresponds to a gap of $E|_{\epsilon=0} = |g|/2$.

## Electronic self-energy

After our simple yet insightful mean-field treatment, we come back to the full model $S'_B$ with four-fermion interactions in Eq. (3) and analyze it in a complementary approach that retains the frequency and momentum dependence. To this end, we employ a matrix-large-$N$ technique similar to[51,52]: we add extra indices to the electrons and bosons, $c_{k,\tau,s} \to c_{k,\tau,s,a}$, $\boldsymbol{d} \to \boldsymbol{d}_{ab}$ and similarly for $\boldsymbol{N}$, where $a, b = 1, 2, \ldots, N$, which are contracted in all terms of $S_B$ so as to ensure $O(N)$ symmetry. In the limit $N \to \infty$, the electronic self-energy $\Sigma$ is given by the "rainbow diagrams"[51,52] shown in Fig. 3(a). In our case, however, $\Sigma$ involves both normal (i.e., particle-number-conserving) and anomalous (non-conserving) contributions as a result of $S_2$ in Eq. (3b). To represent the diagrams algebraically, we again shift to the Bogoliubov-de Gennes basis $\Psi_k = (c_{k,+}, i s_y c^\dagger_{-k,-})^T$, with Pauli matrices $\gamma_i$ acting on this space. In this basis, the free Green's function is $G_0(i\omega, \epsilon) = i\omega - \epsilon \gamma_z$. Up to the first order in $\lambda^2$, see Fig. 3b and Supplementary Appendix B for details on the evaluation of the diagrams, the spin–spin self-energy term can be written as $\Sigma_1(k) = 3\lambda^2 \int_q \frac{\chi_d^{-1}(q)}{2M_q} G_0(i\omega + i\Omega, \epsilon_{\boldsymbol{k}+\boldsymbol{q}})$, while the triplet–triplet term is $\Sigma_2(k) = 12\lambda^2 \int_q \frac{\chi_N^{-1}(q)}{M_q} G_0(i\omega + i\Omega, -\epsilon_{\boldsymbol{k}+\boldsymbol{q}})$. After performing a gauge transformation to make $\phi_0$ real, the anomalous term from the spin–triplet interaction is given by

$$\Sigma_3(k) = 3\phi_0 \int_q \frac{\lambda^2}{M_q} \{\gamma_y, \gamma_z G_0(i\omega + i\Omega, \epsilon_{\boldsymbol{k}+\boldsymbol{q}})\}. \tag{10}$$

For concreteness and since spin fluctuations are believed to occur already at higher energies than superconducting fluctuations in graphene moiré systems[37,38], we focus on $r_d > r_N$; for concreteness, we will use $r_d/r_N = 9$, $v_d^2/v_N^2 = 8$, $\lambda^2/v_N^2\sqrt{r_N} = 1$, $\bar{\chi}_N = \bar{\chi}_d$, and set $\bar{\chi}_\mu = 1$ by rescaling of the fields, but note that no fine-tuning is required for the following results.

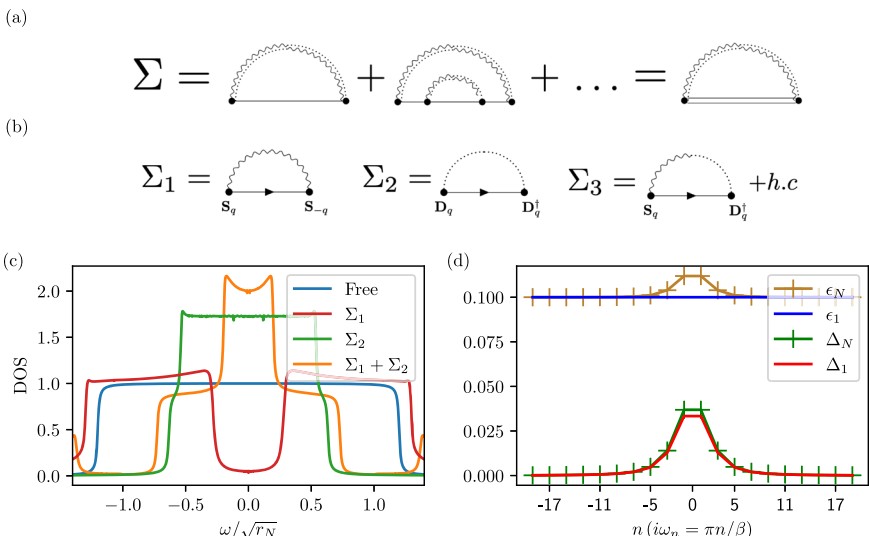

**Fig. 3 | Contributions to self-energy.** Diagrams contributing to the fermionic self-energy $\Sigma$ **a** in the matrix-large-$N$ limit defined in the main text and **b** to first order. **c** Impact of spin ($\Sigma_1$) and triplet fluctuations ($\Sigma_2$) on the constant DOS (blue) of a 2D band with finite bandwidth. (d) Comparing the first order solution ($\epsilon_1, \Delta_1$) and self-consistent solution ($\epsilon_N, \Delta_N$) for $G = i\omega - \epsilon(i\omega)\gamma_z + \Delta(i\omega)\gamma_y$ for $\mathcal{S}_2$ taking into account higher order terms shown in (**a**) (both without momentum integration). We use $\epsilon/\sqrt{r_N} = 0.1, \phi_0/r_N = 0.5$.

## Density of states

Figure 3c shows the effect of the normal contributions of the self-energy $\Sigma_{1,2}$ on the constant DOS around the Fermi level of a 2D band structure. The effect of $\Sigma_1$ is to push the peak of the free spectral function at energy $\epsilon$ away from $\omega = 0$. This results in the opening of a gap (which can be soft depending on the parameter regime), very similar to fluctuating antiferromagnetism discussed in the cuprates[53–55]. $\Sigma_2$ on the other hand, has the opposite effect, where it pushes states toward $\omega = 0$. This is because $\Sigma_1$ and $\Sigma_2$ have the exact same functional form with one key difference: $\epsilon_{\boldsymbol{k+q}}$ of $\Sigma_1$ is replaced by $-\epsilon_{\boldsymbol{k+q}}$ in $\Sigma_2$. As a result, $\Sigma_2$ "sees" the state at energy $-\epsilon$, and pushes it away from 0, resulting in the actual pole at $\omega = \epsilon$ shifting towards $\omega = 0$ (or even crossing 0). This is why high-energy states accumulate in the vicinity of $\omega = 0$.

The effect of the total normal self-energy $\Sigma_1 + \Sigma_2$ is to enhance the DOS in the vicinity of the Fermi level, see Fig. 3c. The anomalous contribution $\Sigma_3$ does not interfere with these effects since it occurs in the $\gamma_y$ channel. The role of $\Sigma_1 + \Sigma_2$ can, thus, be intuitively thought of as providing a renormalized DOS in the normal state, on top of which the anomalous $\Sigma_3$ opens up a gap. We have checked (see Supplementary Appendix C) by numerically solving the self-consistency equation for the self-energy [Fig. 3a] in the limit (of large $v_\mu$) where only the $\boldsymbol{q} = 0$ term of the momentum sum contributes that higher-order corrections do not change our results qualitatively for small $\phi_0$. For instance, Fig. 3d shows the numerical solution for the Green's function $G = i\omega - \epsilon(i\omega)\gamma_z + \Delta(i\omega)\gamma_y$ in Matsubara space upon including the effect of higher-order terms from $\mathcal{S}_2$ [Fig. 3a]; the difference to the first-order result is small.

To gain intuition for the impact of $\Sigma_3$ on the DOS, we first focus again on the $\boldsymbol{q} = 0$ term of the momentum sum in Eq. (10). In this limit, one can easily see (cf. Supplementary Appendix D) that $\Sigma_3$ vanishes linearly in $\epsilon_{\boldsymbol{k}}$ for small energies. Note that this is very different from conventional BCS theory, where the anomalous self-energy is just given by the order parameter $\Delta$ and, thus, constant and finite around the Fermi level; the difference arises from the fact that, although we also keep $\phi_0$ as a constant, it is associated with a four-electron interaction, $\mathcal{S}_2$ in Eq. (3b), and, hence, leads to one- or higher-loop diagrams contributing to $\Sigma_3$. This is what induces the momentum dependence in our case.

Since $\Sigma_3$ is in the $\gamma_y$ channel, the effect of any non-zero value is to generically open a gap. As a result of the linear behavior, $\Sigma_3 \propto \epsilon_{\boldsymbol{k}}$, for $\epsilon_{\boldsymbol{k}} \to 0$, the states exactly at zero energy are unaffected, but slightly away from it, the states get pushed away to higher energy; this is clearly visible in Fig. 4a. The fact that there is still a peak in the spectral function at $\omega = 0$ for finite but sufficiently small coupling strength is in line with previous works on related one-loop self-energy diagrams[56]. In contrast, for large energies, $\Sigma_3$ is readily seen to tend to zero. The spectral function, thus, remains asymptotically unaffected, as can be seen in Fig. 4b. Taken together, we expect the DOS to be reduced (but not fully suppressed for small $\phi_0$) in an energy range around the Fermi level, exhibiting an enhancement with respect to its normal-state value at intermediate energies, and then approaching the normal-state limit at larger energies.

To demonstrate this explicitly beyond the simple $\boldsymbol{q} = 0$ limits, we approximate $\epsilon_{\boldsymbol{k+q}} \simeq \epsilon_{\boldsymbol{k}} + v_F q_\parallel + \boldsymbol{q}^2/(2m)$, where $q_\parallel$ is the component of $\boldsymbol{q}$ along $\boldsymbol{k}$, and numerically evaluate the momentum integrals to find the total self-energy $\Sigma = \Sigma_1 + \Sigma_2 + \Sigma_3$. Choosing $v_F = 1.5v_N, 2m = \sqrt{r_N}/v_N^2$ for concreteness, Fig. 4(c) shows the resulting DOS. As expected, we see that there is a suppression of the DOS. However, for small values of $\phi_0$, the resulting DOS has a V-shaped behavior, which is typically only seen in nodal states (with either nodal lines or points). Recall that the superconducting phase in our model is symmetry-equivalent to a conventional BCS state and that the triplet superconductor that arises at $T = 0$ in-phase (B) will be fully gapped. For larger $\phi_0$, the gap at $\omega = 0$ increases and resembles a hard BCS gap. The suppression of the DOS $\rho_F$ at $\omega = 0$ can be estimated analytically at finite temperature by again taking the limit (of large $v_\mu$) where the integration over $\boldsymbol{q}$ can be replaced by an evaluation at $\boldsymbol{q} = 0$; we find

$$\frac{\rho_F(\phi_0)}{\rho_F(\phi_0 = 0)} = \frac{1}{\sqrt{1+\alpha^2}}, \quad \alpha = \frac{|g|}{4T}, \tag{11}$$

with $g$ defined in Eq. (5). It holds $|\phi_0|^2 < r_d r_N$ due to the Mermin–Wagner theorem. As $|\phi_0|$ increases, $\alpha$ increases the suppression of the DOS, and near the instability point of $|\phi_0|^2 = r_d r_N$, where the bosons would condense, there are no states near the Fermi surface.

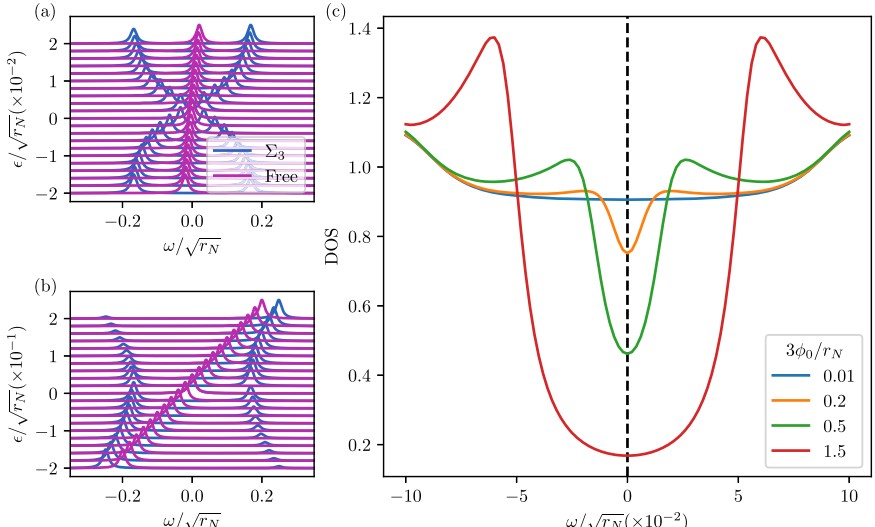

**Fig. 4 | Spectral properties.** Spectral weight as a function of $\omega$ with (blue) and without (purple) $\Sigma_3$ **a** close to $\epsilon_{\boldsymbol{k}} = 0$ and **b** including a larger energy range; in both cases, we focus on the $\boldsymbol{q} = 0$ contribution (see text). **c** The effect of all three self energy contributions $\Sigma_1 + \Sigma_2 + \Sigma_3$ (including the momentum integration) on the DOS. For small $\phi_0$, there is suppression of the DOS at $\omega = 0$, which resembles the V-shaped DOS of a nodal state. For large $\phi_0$, the gap resembles a hard BCS gap.

## Electromagnetic response

We will finally demonstrate that the superconducting phase characterized by $\phi_0 \neq 0$ has the same electromagnetic phenomenology as BCS superconductors, despite the unusual electronic spectral properties. To this end, we study off-diagonal long-range order (ODLRO)[57–59], which implies the Meissner effect[60], flux quantization[61], Josephson effect, and persistent currents[62]. First, focusing on the electrons, we show that $\langle c^\dagger_{s_1,+}(\boldsymbol{x}_1) c^\dagger_{s_2,-}(\boldsymbol{x}_2) c_{s'_2,-}(\boldsymbol{x}'_2) c_{s'_1,+}(\boldsymbol{x}'_1)\rangle \to n_0 (\Psi^*_F(\boldsymbol{x}_{12}))_{s_1,s_2}(\Psi_F(\boldsymbol{x}'_{12}))_{s'_1 s'_2}$, with $\Psi_F \neq 0$, as $|\boldsymbol{x}_j - \boldsymbol{x}'_j| \to \infty$ at finite $\boldsymbol{x}_{12} = \boldsymbol{x}_1 - \boldsymbol{x}_2$ and $\boldsymbol{x}'_{12} = \boldsymbol{x}'_1 - \boldsymbol{x}'_2$, to leading (first) order in $\phi_0$ (see Supplementary Appendix E); as non-zero $\Psi_F$ to linear order in $\phi_0$ implies that it cannot vanish identically for generic $\phi_0$, this is sufficient to show the presence of ODLRO. We find the "macroscopic wave function" to be a singlet, $\Psi_F(\boldsymbol{x}) = is_y \psi_F(\boldsymbol{x})$, as expected since spin-rotation symmetry is preserved at finite $T$, with $\psi_F(\boldsymbol{x})$ shown in Fig. 5a. Alternatively, one can demonstrate ODLRO to arbitrary order in $\phi_0$, by focusing on the bosons: to zeroth order in $\lambda$, we find $\langle(\boldsymbol{d}^\dagger(\boldsymbol{x}_1)\boldsymbol{N}(\boldsymbol{x}_2))(\boldsymbol{d}(\boldsymbol{x}'_1)\boldsymbol{N}(\boldsymbol{x}'_2))\rangle \to \psi^*_B(\boldsymbol{x}_{12})\psi_B(\boldsymbol{x}'_{12})$ as $|\boldsymbol{x}_j - \boldsymbol{x}'_j| \to \infty$, with $\psi_B(\boldsymbol{x})$ plotted in Fig. 5b along with an analytic asymptotic form for large $\boldsymbol{x}$; in Supplementary Appendix F, we show that this leads to the same constraints as the conventional form of bosonic ODLRO[57,58].

Finally, the connection to the textbook theory of superconductivity can be made more explicit by deriving the analog of the time-dependent Ginzburg–Landay theory: we reinstate fluctuations via $\phi_0 \to \phi(\boldsymbol{x}, \tau)$ and integrate out all other degrees of freedom yielding

$$\mathcal{S}_{\mathrm{GL}} = \int_{\boldsymbol{x},\tau}\left[\rho|D_\tau\phi|^2 + (r_\phi + |c_1|^{-1})|\phi|^2 + v^2\,|\boldsymbol{D}\phi|^2\right] \quad (12)$$

to leading order in $\phi$ and gauge-covariant derivatives $(D_\tau, \boldsymbol{D})_\mu = \partial_\mu - i2eA_\mu$. We evaluated the coefficients in $\mathcal{S}_{\mathrm{GL}}$ to leading (zeroth) order in $\mathcal{S}_c$ and find $\rho, v_\phi > 0$ and $r_\phi < 0$ for low $T$ (see Supplementary Appendix G); the state with QLRO in $\phi_0$ thus corresponds, as usual, to the Higgs phase, with Meissner effect and massive Higgs mode, but without Goldstone modes.

## Discussion

We have studied the finite-$T$ vestigial phases, see Fig. 1, associated with two primary order parameters, $\boldsymbol{d}$ and $\boldsymbol{N}$, describing a fully gapped triplet superconductor and spin magnetism, respectively. While this includes phases where only broken time-reversal

symmetry (C) or charge-$4e$ pairing (phase A) can survive finite-temperature fluctuations, our main focus has been on phase $B_{1,2}$, which is best thought of as a condensate of spin-0 bosons $\phi$ formed by two electrons in a triplet state and an electron and a hole, which are also in a triplet configuration. This defines a Higgs phase without any broken symmetry, effective time-dependent Ginzburg–Landau-like action given by Eq. (12), and ODLRO; meanwhile, its spectral electronic properties are very unusual: as can be seen in Fig. 2(d) and Fig. 4(c), varying $\langle\phi\rangle = \phi_0$ changes the low-energy DOS from partial suppression, akin to that of a nodal superconducting state, to a hard gap.

We emphasize that such a state is generically expected to emerge in a range of small but finite temperatures in a two-dimensional system that exhibits triplet superconductivity and spin magnetism at $T = 0$. The current experiment indicates that alternating-twist-angle graphene moiré superlattices indeed realize this phenomenologically and, thus, provide an ideal testbed for this theory. In fact, recent tunneling data in the superconducting state[34,35] show a V-shaped DOS that can become U-shaped upon doping. While this could also be explained by an exotic interband order parameter[63], our proposed state provides a very natural interpretation of these observations: the value of $\phi_0$ is expected to vary from sample to sample and also change with electron filling, likely decreasing when moving further away from the insulator, which would lead to a transition from U- to V-shaped, as seen in experiment[34]. We finally point out that the suppression of $\boldsymbol{N}$ would immediately also suppress $\phi_0$ in our model and could, therefore, explain why superconductivity is connected to the reset behavior in trilayer graphene[34,35,39,40].

Naturally, there are many open questions related to the novel state we propose here, such as understanding quantitative differences in its electromagnetic response, such as penetration depth, compared to a conventional BCS state and its behavior under the influence of disorder. Furthermore, it would be interesting to study the exotic spectral properties with other techniques such as Monte-Carlo simulations and dynamical mean-field theory. We leave this for future work.

## Methods
### Mean-field form of the bosonic interactions
In the discussion above, we viewed the field theory defined by the action $\mathcal{S} = \mathcal{S}_e + \mathcal{S}_\chi + \mathcal{S}_c + \mathcal{S}_V$ as an effective low-energy theory that arises when high-energy electronic degrees of freedom have already been

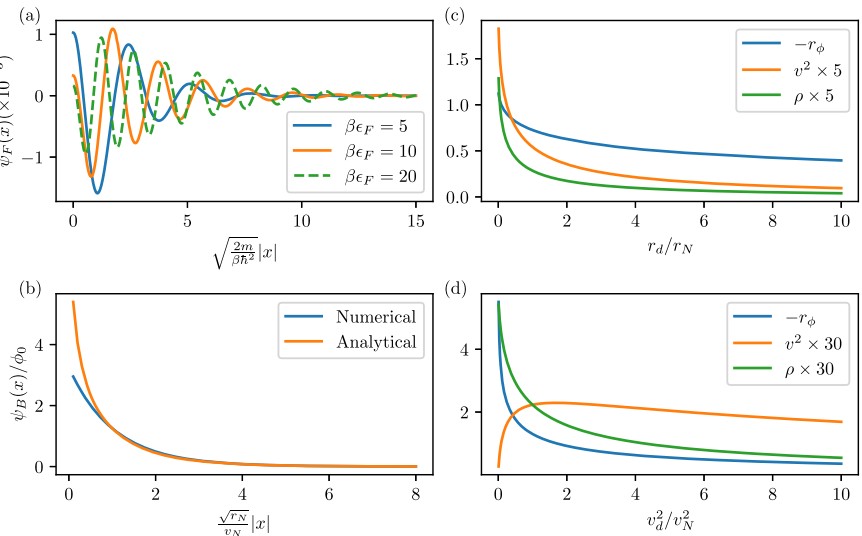

**Fig. 5 | ODLRO and electromagnetic response. a** The fermionic and **b** the bosonic ODLRO "macroscopic wavefunction". The mass $r_\phi$ [in units of $r_N^{-1/2} v_N^{-2}$], superfluid density $\rho$ [$r_N^{-3/2} v_N^2$], and velocity $v^2$ [$r_N^{-3/2}$] of $\mathcal{S}_{GL}$ in Eq. (12) as a function of $r_d$ and $v_d^2$ are shown in (**c**) and (**d**), respectively.

integrated out. Due to the symmetry and locality constraints, it only depends on a few parameters, $r_\mu$, $v_\mu$, $b_{1,2,3}$, $c_{1,2}$. As can be seen in Fig. 1(c), in particular, (the sign of) the parameters $c_1$ and $b_2$ entering $V$ crucially determine the phase of the system. We here provide an estimate for these parameters by computing them from high-energy electronic degrees of freedom within the mean-field theory. To this end, we replace the bosonic fields with classical homogeneous and time-independent vectors, $\boldsymbol{N}_q \to \delta_{q,0}\boldsymbol{N}$, $\boldsymbol{d}_q \to \delta_{q,0}\boldsymbol{d}$, in $\mathcal{S}_e + \mathcal{S}_\chi + \mathcal{S}_c$; this yields

$$\mathcal{S}_{HE} = \int_k c_{k,\tau,s}^\dagger (-i\omega_n + \epsilon_{\tau \cdot k}) c_{k,\tau,s}$$
$$+ \lambda \int_k \left[ c_k^\dagger \boldsymbol{s} \cdot \boldsymbol{N} \tau_z c_k + \left( c_k^\dagger \boldsymbol{s} \cdot \boldsymbol{d} \, i s_y \tau_y c_{-k}^\dagger + \text{H.c.} \right) \right], \quad (13)$$

which we now view as our full action, also containing the high-energy degrees of freedom. Integrating out the electronic degrees of freedom and expanding the resulting action in terms of $\boldsymbol{N}$ and $\boldsymbol{d}$ to quartic order, one obtains exactly the same terms as in $V$ in Eq. (1), as expected by symmetry. Moreover, one finds

$$c_1 = b_2 = -b_1/2 < 0, \quad (14)$$

$$\text{with} \quad b_1 = 32\lambda^4 T \sum_{\omega_n} \int \frac{d^2 \boldsymbol{k}}{(2\pi)^2} \frac{1}{(\omega_n^2 + \epsilon_k^2)^2} > 0. \quad (15)$$

As stated in the main text, this places us into phase (B). We note, however, that fluctuation corrections to the mean field can modify the values of these coupling constants significantly[46,64,65]. For instance, ferromagnetic fluctuations can change the sign of $b_2$ to positive values[46].

## Data availability
The data generated in this study are available in the Zenodo database under the accession code https://zenodo.org/records/10547103.

## Code availability
The codes used to generate the plots are available from the corresponding author upon request.

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

## Acknowledgements
We thank Rafael Fernandes, Victor Gurarie, Peter Orth, and Subir Sachdev for fruitful discussions on the project and Jakob Wessling for a related collaboration. M.S.S. acknowledges funding from the European Union (ERC-2021-STG, Project 101040651—SuperCorr). Views and opinions expressed are, however those of the authors only and do not necessarily reflect those of the European Union or the European Research Council Executive Agency. Neither the European Union nor the granting authority can be held responsible for them. P.P.P. acknowledges support from the Laboratory for Physical Sciences through the Condensed Matter Theory Center.

## Author contributions
P.P.P. and M.S.S. performed the research and wrote the paper.

## Funding

## Competing interests
The authors declare no competing interests.
