## [Peer Review File · Nature Communications]

Vestigial singlet pairing in a fluctuating magnetic triplet superconductor and its implications for graphene superlatticesREVIEWER COMMENTS

Reviewer #1 (Remarks to the Author):

This manuscript, authored by P.P. Poduval and M.S. Scheurer, is inspired by experiments on twisted stacked layers of graphene. It studies a phenomenological model of a material capable of exhibiting triplet-superconducting (d) and magnetic (N) order parameters. The action consists of the action of free electrons, the Ornstein-Zernicke-type quadratic actions of d and N , and the most generic quartic term incorporating five phenomenological constants.

I stress the phenomenological character of the action. In principle, superconducting and magnetic order parameters are made of electronic degrees of freedom, with microscopic descriptions usually either in terms of these electronic degrees or the magnetic and superconducting order parameters after integrating out fermions. In contrast, this manuscript's action encompasses all of these degrees of freedom (electrons + superconducting order + magnetic order). Although phenomenological, such approaches are not uncommon in the area of strongly correlated systems, exemplified by the spin-fermion model.

At the mean-field level, the model exhibits a phase labeled as "phase B," characterized by the order parameter $\vec{\phi} \sim \vec{d} \cdot \vec{N}$. This phase is one of several defined by their superconducting and magnetic order parameters. Subsequently, the authors explore an effective action wherein ϕ interacts with fluctuations of d and N , facilitating the study of the electron density of states (DoS). The DoS is evaluated perturbatively by computing electron self-energy due to interactions with these fluctuations and by solving associated self-consistency equations. The self-consistent approach is valid in the limit of an infinite number of electronic flavors (a minor point worth noting: self-consistency equations are generically controlled by large numbers of flavors; crediting Ref. 50 for this is somewhat unusual) but, technically speaking, is uncontrolled for real systems.

The calculations culminate in finding the suppression of the DoS at the Fermi energy. Depending on the parameters of the model, it ranges from partial suppression, designated as the "V-shaped DoS," to complete vanishing, referred to as the "U-shaped DoS." The authors argue that the computed V- and U-shaped DoSs align with the recent experimental results from Ref. 34.

This is a high-quality manuscript with meticulous, comprehensive calculations. The results are not uninteresting and are possibly of experimental relevance. However, I think this manuscript should find home in a journal like Physical Review B, as opposed to the broader purview of Nature Communications. My main criticism is the manuscript's lack of sharp, reliable results of sufficient significance for Nature Communications.

Indeed, the idea of phase B appears pedestrian once the possibility of both the spin-triplet parameter d and the magnetic parameter N is assumed, and the action obviously allows for that phase in some range of parameters, at least at the mean-field level. The existence of phase B is virtually postulated in the manuscript and not derived.

I am also sceptical that the manuscript explains the experimentally observed V- and U-shaped DoSs. The issue is that DoS suppression at the Fermi energy is a common property of metallic systems with repulsive interactions and can come from various types of interactions. For example, Coulomb interactions always yield singular negative corrections to the DoS in metals (this is not to say that the observed gaps in trilayer graphene come from Coulomb interactions, but to stress how common this is qualitatively). In narrow-band 2D materials, such corrections will be sensitive to the doping level. The manuscript's implication linking the experimentally observed DoS gaps to the order parameter ϕ lacks a compelling basis.

(A minor nomenclature remark: the authors apparently call the full DoS suppression at some energy a "hard gap". In the literature, the meaning of a hard gap is usually different: it means the vanishing of the DoS in a finite energy interval.)

Lastly, it is essential to acknowledge that the calculations in this work rely on uncontrolled approximations, combining elements of self-consistency, mean-field theory, and the initial phenomenological model.

In conclusion, the manuscript offers valuable insights and meticulous calculations but, I think, falls short of reaching the threshold for publication in Nature Communications. Instead, I recommend resubmission to a journal like Physical Review B.

Reviewer #2 (Remarks to the Author):

The key result of this article is the unveiling of intertwined of vestigial order between triplet superconductivity and antiferromagnetic order, which is highly likely to be relevant for as yet unexplained experimental observations in bilayer graphene.

This work (and I would venture follow-ups thereof by the authors or others) are likely to be of prominent significance in this field. As the authors describe, bilayer graphene systems show a very rich variety of simple and complex phases and many of the latter could be of the intertwined type where this type of analysis applies.

The methodology is sound, and I am confident the analysis is correct, but I must confess that in the version as written now, this is hard to follow or to reproduce. The authors tried to be concise, but in my view this has come at the price that the main story is often lost.

Specifically:

1. The $T=0$ analysis and $T>0$ analysis are simultaneously discussed. Often one has to think three times whether a statement belongs to the one or the other. The main story w.r.t. experiment has to be the finite T analysis, and for a Nature Comm. type audience I would recommend presenting $T>0$ first and discuss $T=0$ either summarily at the end of the section or in the appendix.
2. Fig 1. has many symbols that are only explained later.
3. At the presentation level, it appears the microscopic ($S_e+S_c+S_{int}$) (what they call S_{HE} in the supp. mat., though without S_{χ}) are mixed with the mean field action ($S_{\chi+V}$).
4. In the mean field analysis the role of the (sign) of the parameters in relation to the groundstate would help to understand the symmetry breaking patterns.
5. The role of the parameter r_{μ} is very unclear. Shouldn't this be $r_{\mu}=1$ by the definition of χ^0 .
6. The section on the Electronic self energy is very confusing:
 - 6.a) In Eq.2 the fields S, D are sources for N, d so the susceptibility rather than the inverse susceptibility should appear.
 - 6.b) Though clearly there is a limit where in the four-Fermi theory $S_e+S_1+S_2$ the double fermion internal loop may be approximated by a single composite effective $N(S)$ or $D(d)$ line, it is not clear what this regime is. This should be expressly stated.
 - 6.c) It is unclear what is meant by "anomalous contributions".
 - 6.d) Fig2d the term "self-consistent solution" is unexplained.
 - 6.e.) The paragraph "Before proceeding" appears to be a computational detailed discussion best deferred to supp.mat. This is clearly illustrated by the use of the identical sentence "The anomalous contribution ... channel" in both the preceding and subsequent paragraph.
 - 6.f) Though the analysis of the DOS should be there for the experimental reason highlighted by the authors, the section only gives mathematical explanations, but no physical ones for this effect.
7. In that sense the minimal mean field theory model is much more instructive.

I would urge the authors to revise and rewrite with the broader Nat.Comm. audience in mind, and resubmit.

Reviewer #3 (Remarks to the Author):

This MS presents a comprehensive study of vestigial order in a 2D system with attraction in valley FM/inter-valley AMF channel and valley-singlet spin-triplet superconducting channel. Neither AFM nor B-type spin-triplet superconductivity are possible in 2D at a finite T , however composite (vestigial) orders are potentially possible as they do not break $O(3)$ continuous symmetry. Previous works on vestigial orders focused primarily on the superconducting channel and argued that such order can be viewed as $4e$ superconductor. The authors analyzed in great detail different composite order - the one with the charge $2e$. They demonstrated that such an order is a spin-singlet 4-fermion bound state between a spin-triplet particle-particle pair and a $S=1$ soft particle-hole excitation associated with valley FM (3 electrons and one hole or 3 holes and one electron). They argued that the charge $2e$ bound state possesses quasi-long-range order at a finite T in 2D and displays a Meissner effect, like a conventional spin-singlet superconductor, but the suppression of the density of states at small frequencies is more complex than in a conventional case. The authors argue that their results are relevant to experiments on twisted bilayer graphene.

I truly enjoyed reading this paper. It is of high-quality, and the authors presented enough details of their calculations, so a reader can follow their analysis. My recommendation is to accept this MS to Nature Communications.

Two suggestions:

1. On the spin-triplet/valley singlet order with a constant gap. The authors probably meant the analog of a B-type phase for $3He$, not the A-type phase. I would suggest to add an explanatory phrase on this.
2. About the one-loop expressions for the self-energy and the discussion after Eq. (3). I suggest to state explicitly that the peak in the spectral function is pushed away from $\omega=0$ only if the coupling λ exceeds a certain limit.

Point-by-point response to referee 1:

This manuscript, authored by P.P. Poduval and M.S. Scheurer, is inspired by experiments on twisted stacked layers of graphene. It studies a phenomenological model of a material capable of exhibiting triplet-superconducting (d) and magnetic (N) order parameters. The action consists of the action of free electrons, the Ornstein-Zernicke-type quadratic actions of d and N , and the most generic quartic term incorporating five phenomenological constants.

I stress the phenomenological character of the action. In principle, superconducting and magnetic order parameters are made of electronic degrees of freedom, with microscopic descriptions usually either in terms of these electronic degrees or the magnetic and superconducting order parameters after integrating out fermions. In contrast, this manuscript's action encompasses all of these degrees of freedom (electrons + superconducting order + magnetic order). Although phenomenological, such approaches are not uncommon in the area of strongly correlated systems, exemplified by the spin-fermion model.

Response: We thank the referee for their time and report. And, yes, we agree that the theory we study is very similar in spirit to (yet different of course from) the spin-fermion model, a widely studied and very successful description of complex quantum materials. We have added a comment on this relation right in the beginning of the section “Model” in the revised version of the manuscript.

At the mean-field level, the model exhibits a phase labeled as “phase B,” characterized by the order parameter $\phi \sim \mathbf{d} \cdot \mathbf{N}$. This phase is one of several defined by their superconducting and magnetic order parameters. Subsequently, the authors explore an effective action wherein ϕ interacts with fluctuations of d and N , facilitating the study of the electron density of states (DoS). The DoS is evaluated perturbatively by computing electron self-energy due to interactions with these fluctuations and by solving associated self-consistency equations. The self-consistent approach is valid in the limit of an infinite number of electronic flavors (a minor point worth noting: self-consistency equations are generically controlled by large numbers of flavors; crediting Ref. 50 for this is somewhat unusual) but, technically speaking, is uncontrolled for real systems.

Response: We thank the referee for their remark on Ref. 50. We would like to clarify that Ref. 50 is credited *not* for the analysis of the density of states as outline above, but for arriving at the effective theory \mathcal{S}_B , with crucial term \mathcal{S}_ϕ in Eq. (1). Also, in that case, N is *not* the number of electronic flavors but the number of components of \mathbf{N} and \mathbf{d} —this is an important distinction since the former would lead to the usual mean-field theory which would violate Mermin-Wagner but the latter, which we use, does not; we emphasize this

more strongly in the revised version of the manuscript. We decided to cite Ref. 50 in that context as they also perform such a large- N analysis in a setting that is most similar to what we have (vestigial order, albeit a rather different form of it). However, if the referee has a particular paper in mind we should also cite, we are happy to do that. Concerning the comment that that large- N is ultimately uncontrolled since the real system has $N = 3 < \infty$, we note that we only use it to formally justify \mathcal{S}_B while, at the same time, \mathcal{S}_B can be seen as the simplest effective field-theory for phase (B). Since the existence of phase (B) in the phase diagram for a system with fluctuating triplet superconductivity and spin magnetism, like graphene moiré systems, is a very natural (“pedestrian” to quote the referee below) consequence of the Mermin-Wagner theorem, \mathcal{S}_B is an extremely plausible effective theory, beyond the large- N calculation.

The calculation of the density of states is not done using the abovementioned large- N approach. In fact, as we detail further in the response to the next question of referee, we use three different, yet complementary theoretical approaches all yielding qualitatively consistent results for the spectral properties in the superconductor. This makes us confident that our results for the density of states are reliable.

The calculations culminate in finding the suppression of the DoS at the Fermi energy. Depending on the parameters of the model, it ranges from partial suppression, designated as the “V-shaped DoS,” to complete vanishing, referred to as the “U-shaped DoS.” The authors argue that the computed V- and U-shaped DoSs align with the recent experimental results from Ref. 34.

This is a high-quality manuscript with meticulous, comprehensive calculations. The results are not uninteresting and are possibly of experimental relevance. However, I think this manuscript should find home in a journal like Physical Review B, as opposed to the broader purview of Nature Communications. My main criticism is the manuscript’s lack of sharp, reliable results of sufficient significance for Nature Communications.

Response: First, we are happy to hear that the referee judges our manuscript to be “a high-quality manuscript with meticulous, comprehensive calculations” and that the results are “not uninteresting and are possibly of experimental relevance”. At the same time, we are surprised to hear that the referee still concludes that there is a lack of “sharp, reliable results of sufficient significance for Nature Communications” and we strongly disagree with this conclusion. While we will address the remaining more specific questions of the referee further below, we next explain why our conclusions are reliable and of high significance.

Reliable. As already stated above we do *not* use the large- N approach of Ref. 50 to compute the density of states (one of our central results). Instead, we use in total three different, complementary approaches which

yield consistent results:

- (i) We compute the self-energy to leading order in the coupling constant λ . This approach has the advantage of allowing us to keep the full frequency dependence and perform the analytic continuation ‘by hand’. Of course, it has the weakness of only applying to the limit of small λ . While this is in fact the regime with the partial suppression/V-shaped DOS we are most interested in and thus not that limiting for our purposes, one might be worried that there are additional non-perturbative solutions.
- (ii) For this reason, we supplement this with a non-perturbative approach, controlled in a *matrix-large- N* limit (also used in Phys. Rev. B **89**, 165114 in a different context), where all rainbow diagrams are summed up. We emphasize that this is *not* a (conventional) mean-field approximation and keeps the full frequency dependence. We checked that for a large range of couplings (including those in Fig. 3c) we do not find any additional non-perturbative solution and even good quantitative agreement with the perturbative approach (see Appendix D for numerics and Appendix H.2 for an analytic demonstration in a special limit). This justifies our computation of the DOS shown in Fig. 3(c).
- (iii) To test our approach even further, we have also performed a self-consistent mean-field treatment of the spectral properties of the superconductor. Remarkably, it reveals the same behavior of the DOS: for small coupling, we just get partial suppression at low-energies and a V-shape, see Fig. 4.

We hope that this could clarify our approach to the referee. The fact that we use three different complementary approaches yielding the same behavior should remove any doubt that our computations are reliable.

Significance. We next elaborate on our justification for publication in Nature Communications:

- (I) While the BCS theory of superconductivity is characterized by the condensation of a two-electron operator that transforms trivially under all symmetries of the system, known extensions of this “conventional” form of pairing are “unconventional superconductors” where the two-electron operator transforms under a non-trivial irreducible representation of the system’s point group and charge- $4e$ (or even higher-charge) superconductors where a four-electron operator condenses. In our manuscript, we introduce an additional novel type of superconducting state, distinct from all those mentioned above: like the BCS state, the elementary operator that condenses transforms trivially under all symmetries of the normal state. It is also a charge- $2e$ object, however, it consists of four operators—three electrons and a hole. We develop and study a systematic theory for such a phase, which shows that its spectral properties are rather different from those of the BCS state: depending on temperature and

the strength of superconductivity, the electronic spectrum can be either fully gapped (as in BCS theory) or look more like that of an unconventional nodal or gapless superconductor. Yet, we show that it exhibits the conventional superconducting phenomenology of Meissner effect and dissipationless current.

- (II) Our analysis also shows that such a state is not a fine-tuned phenomenon but rather expected to occur generically in a 2D system with strong tendencies towards both spin magnetism (order parameter N) and triplet superconductivity (d): the Mermin-Wagner theorem does not allow for either of these orders at finite temperature, however, their product, $d \cdot N$, can order (more precisely: exhibit quasi-long-range order), defining the novel superconducting state described above, which can hence be thought of as a “vestigial phase” of N and d . As such, our work also introduces a novel state to the very active field of vestigial order.
- (III) While we emphasize that the generality of the phenomenon and our general discussion of it is a strength rather than a weakness of our work, graphene moiré systems are (to the best of our knowledge and as of now) the most natural class of systems to expect this pairing state to occur: they exhibit strong ordering tendencies to both triplet superconductivity and magnetism, which coexist in some part of the phase diagram. As such, our work connects and significantly extends two very active research directions, vestigial phases and graphene moiré superlattices.
- (IV) Furthermore, it makes an important step towards solving (or might even have solved) important challenges in the field of graphene moiré systems—understanding the recent puzzling tunneling experiments showing V-shaped or U-shaped tunneling data (depending on sample and filling) and determining the type of superconductivity realized in these systems: our novel pairing is not only naturally expected to occur in these systems but also captures the tunneling phenomenology, without having to postulate additional transitions etc.. This can be seen as both an independent confirmation of our state and as a natural explanation of the experiment. It also makes the for the field of graphene moiré systems (and beyond) important point that the tunneling data does *not* imply a nodal state.

In summary, in our opinion, the work opens up exciting new avenues in the field of superconductivity (see I), in the field of vestigial phases (see II), and in graphene moiré systems (see III) by introducing a novel type of pairing state, that can be thought of as a vestigial of magnetism and triplet superconductivity that is generically expected to occur in a 2D system, like graphene moiré systems, with a tendency towards both of these zero-temperature orders. Furthermore, it also provides a solution to the critical open question of how to understand the transition from V-shape to fully gapped in tunneling data in graphene moiré systems

(see IV). In combination with the fact that it is relevant to and expected to lead to significant attention across several very active subfields of physics—moiré superlattices, fundamentals of superconductivity, and vestigial phases—our manuscript seems very well suited for publication in Nature Communications. This also seems to be in line with the conclusions of the other referees.

Indeed, the idea of phase B appears pedestrian once the possibility of both the spin-triplet parameter d and the magnetic parameter N is assumed, and the action obviously allows for that phase in some range of parameters, at least at the mean-field level. The existence of phase B is virtually postulated in the manuscript and not derived.

Response: First, we agree with the referee that the phase B in the system is expected to arise rather generally in a system with strong tendencies towards both spin magnetism and triplet superconductivity—this is clearly a strength rather than a weakness. As already noted above and in the introduction, graphene moiré systems like twisted bi- and mirror-symmetric tri-layer graphene provide a very natural and timely example of systems where experiments (rather than any form of ultimately uncontrolled theoretical calculations) indicate tendencies to both spin magnetism and triplet superconductivity. We further would like to add that we showed that any such system not only just contains phase (B) somewhere in parameter space/in the phase diagram but, out of all phases in the phase diagram in Fig. 1, phase (B) is generically expected to be favored [we computed the coefficients $b_{1,2,3}$ and $c_{1,2}$ which placed us in the region (B)]. Furthermore, we emphasize that once a phase with $d \cdot N \neq 0$ is realized at $T = 0$, general arguments based on Mermin-Wagner imply that our finite- T phase (B) has to exist in a finite range of temperatures above $T = 0$.

I am also sceptical that the manuscript explains the experimentally observed V- and U-shaped DoSs. The issue is that DoS suppression at the Fermi energy is a common property of metallic systems with repulsive interactions and can come from various types of interactions. For example, Coulomb interactions always yield singular negative corrections to the DoS in metals (this is not to say that the observed gaps in trilayer graphene come from Coulomb interactions, but to stress how common this is qualitatively). In narrow-band 2D materials, such corrections will be sensitive to the doping level. The manuscript's implication linking the experimentally observed DoS gaps to the order parameter ϕ lacks a compelling basis.

Response: We thank the referee for their comments. Indeed, the referee is correct that the suppression of the DoS at the Fermi energy is a generic property in interacting metallic systems. However, we would like to emphasize that we are not in a metallic system, but in a superconducting system which changes the story dramatically.

In typical theories of superconductivity, the fermionic spectrum is either fully gapped, like in a BCS-

type s -wave state, with a U-shaped DoS, or nodal (e.g., in a p -wave state) with a V-shaped DoS. However, recent experiments on twisted graphene systems [Nature **606**, 494 (2022)] shows a situation where the superconductor exhibits a crossover from U-shaped DoS to a V-shaped DoS. As far as we are aware, the experimentalists [also in the related work Nature **600**, 240 (2021)] are confident that the data is taken in the superconducting state.

It is unclear what causes a crossover from nodal to fully gapped DoS in these systems, but the answer will lead to a higher understanding of the underlying intriguing phenomena. In this work, we provide one possible and not implausible answer to this question, which is central to the field.

(A minor nomenclature remark: the authors apparently call the full DoS suppression at some energy a “hard gap”. In the literature, the meaning of a hard gap is usually different: it means the vanishing of the DoS in a finite energy interval.) Lastly, it is essential to acknowledge that the calculations in this work rely on uncontrolled approximations, combining elements of self-consistency, mean-field theory, and the initial phenomenological model.

Response: We would like to clarify that we do indeed find a hard gap, where the DoS vanishes in a finite energy interval. One of the calculations we performed was a fully self-consistent calculation over an effective Hamiltonian interaction, which shows a transition from a partial suppression of DoS to a fully gapped state, see Fig. 4d. Regarding the accuracy of our approximations, we re-iterate our previous response above that we have verified our calculations using three independent methods, all agreeing with each other qualitatively.

In conclusion, the manuscript offers valuable insights and meticulous calculations but, I think, falls short of reaching the threshold for publication in Nature Communications. Instead, I recommend resubmission to a journal like Physical Review B.

Response: We hope that we have convinced the referee with our response that the work does satisfy the criteria of Nature Communications.

Point-by-point response to referee 2:

The key result of this article is the unveiling of intertwined of vestigial order between triplet superconductivity and antiferromagnetic order, which is highly likely to be relevant for as yet unexplained experimental observations in bilayer graphene.

This work (and I would venture follow-ups thereof by the authors or others) are likely to be of prominent significance in this field. As the authors describe, bilayer graphene systems show a very rich variety of simple and complex phases and many of the latter could be of the intertwined type where this type of analysis applies.

The methodology is sound, and I am confident the analysis is correct, but I must confess that in the version as written now, this is hard to follow or to reproduce. The authors tried to be concise, but in my view this has come at the price that the main story is often lost.

Response: We thank the referee for their time and helpful report; we are also happy to read that they judge our work “highly likely to be relevant for as yet unexplained experimental observations in bilayer graphene” and “likely to be of prominent significance in this field”.

We agree with the referee’s statement that our presentation in the previous version of the manuscript was in certain parts a bit too concise. As we detail below, we have taken their inputs into consideration, which has substantially improved our work, in particular, making it more accessible to the reader.

Specifically: 1. The $T=0$ analysis and $T>0$ analysis are simultaneously discussed. Often one has to think three times whether a statement belongs to the one or the other. The main story w.r.t. experiment has to be the finite T analysis, and for a Nature Comm. type audience I would recommend presenting $T>0$ first and discuss $T=0$ either summarily at the end of the section or in the appendix.

Response: We agree with the referee that the $T = 0$ and $T > 0$ analyses should be clearly separated. We made sure that this is the case in the revised manuscript by including a dedicated section “Zero-temperature phases”. We emphasize that the vestigial phases at $T > 0$ follow directly from general symmetry considerations from the $T = 0$ analysis such that the most intuitive motivation and explanation of our finite- T analysis has to be introduced starting at $T = 0$. However, with the additional subsection and the modified explanations in the second and third subsection of “Results”, it should become very clear to the reader that anything after and including the new subsection “Vestigial phases at finite T ” refers to $T > 0$.

2. *Fig 1. has many symbols that are only explained later.*

Response: Indeed, in the previous version of the manuscript, we already referred to Fig. 1 in the introduction which implied that all symbols in Fig. 1(a) should be accessible to reader when only reading the introduction. This is of course not the case since many symbols have not been introduced. To fix this in the revised version, we interchanged the order of panels (a) with (b,c) and made sure that all symbols used in current Fig. 1(c) [previously Fig. 1(a)] are explained in the text in detail when we refer to it and discuss it for the first time (now in the results section).

3. *At the presentation level, it appears the microscopic ($S_e + S_c + S_{int}$) (what they call S_{HE} in the supp. mat., though without S_χ) are mixed with the mean field action ($S_\chi + V$).*

Response: We agree with the referee that—as a natural consequence of the complexity of the problem we study and the fact that we use different approaches to address it—there are several different combinations of actions that appear in our analysis. To clarify and summarize, we consider the following different actions:

1. In the supplement we use the “high-energy” action \mathcal{S}_{HE} to derive expressions for the parameters $b_{1,2,3}$ and $c_{1,2}$ in the
2. “effective low-energy action” we start our analysis with in the main text; it is given by $\mathcal{S}_e + \mathcal{S}_c + \mathcal{S}_\chi + \mathcal{S}_V$.
3. To identify the possible phases at zero temperature, which are listed in the phase diagram of Fig. 1, one simply minimizes the potential V in \mathcal{S}_V .
4. Phase (B) is described by the action $\mathcal{S}_B = \mathcal{S}_e + \mathcal{S}_c + \mathcal{S}_\chi + \mathcal{S}_\phi$, where the new contribution \mathcal{S}_ϕ is defined in Eq. (1).
5. Integrating out the bosonic fields in \mathcal{S}_B yields a purely electronic action for phase (B), given by $\mathcal{S}'_B = \mathcal{S}_e + \mathcal{S}_{int}$, where \mathcal{S}_{int} is defined in Eq. (2).

To make this more transparent to the reader, we have revised our discussion in several places in the manuscript [we mention explicitly in “Zero-temperature phases” that \mathcal{S}_V is minimized to arrive at Fig. 1c, introduced the more intuitive notation \mathcal{S}_B and \mathcal{S}'_B in “Theory for phase (B)”, explain more clearly that the action contributions in Eq. (2) are four-fermion interactions, and changed the notation in the section “Minimal mean-field theory” to highlight the differences to and the connections with other parts of the paper] to define more clearly what the precise form of the action is that is studied in the respective paragraph.

4. *In the mean field analysis the role of the (sign) of the parameters in relation to the groundstate would help to understand the symmetry breaking patterns.*

Response: This is a great suggestion since what type of ground state is favored by the different terms in $V = b_1(\mathbf{d}^\dagger \mathbf{d})^2 + b_2|\mathbf{d}\mathbf{d}|^2 + b_3\mathbf{N}^4 + c_1|\mathbf{d}\mathbf{N}|^2 + c_2(\mathbf{d}^\dagger \mathbf{d})\mathbf{N}^2$ is very intuitive: while b_1, b_3, c_2 primarily control the magnitude of the different components and the stability of the field theory, the key players are b_2 and c_1 —this is, by the way, the reason why we use b_2 and c_1 on the axis of the phase diagram in Fig. 1(c) [previously Fig. 1(a)]. Negative (positive) c_1 favors (disfavors) \mathbf{N} and \mathbf{d} to have parallel components, which is why the phases with finite ϕ_{dN} appear for $c_1 < 0$ in the phase diagram. Similarly, negative (positive) b_2 favors (disfavors) ϕ_{dd} which clearly consistent with the phase diagram, too. We have added comments on this in the new section “Zero-temperature phases” in the revised version of the manuscript.

5. *The role of the parameter r_μ is very unclear. Shouldn't this be $r_\mu = 1$ by the definition of χ^0 .*

Response: In our notation, the susceptibilities (spin and superconducting for $\mu = N$ and $\mu = d$, respectively) or, equivalently, the propagators of the bosons are parametrized as $\chi_\mu(q) = \chi_\mu^0/(r_\mu + \Omega_n^2 + v_\mu^2 q^2)$; upon analytical continuation, $\Omega_n^2 \rightarrow -(\Omega + i0^+)^2$, it becomes clear that the gap of \mathbf{N} and \mathbf{d} is given by r_μ . Note that we used χ_μ^0 to set the prefactor of Ω_n^2 in the denominator of $\chi_\mu(q)$ to 1.

6. *The section on the Electronic self energy is very confusing:*

6.a) *In Eq.2 the fields S, D are sources for N, d so the susceptibility rather than the inverse susceptibility should appear.*

Response: We would like to clarify that the $M_q = \chi_d^{-1}\chi_N^{-1} - |\phi_0|^2$ factor in the action is the determinant of the coupling matrix of the bosonic fields, which contains the product of both (inverse) susceptibilities. This is why in the way we have presented the self-energy, it is the inverse susceptibility that appears. Indeed, in the limit $\phi_0 \rightarrow 0$, we can see that the determinant term cancels with the inverse-susceptibility, $\chi_{d,N}^{-1}/M_q \rightarrow \chi_{N,d}$, and the result will be as the referee correctly expects. As this can be confusing, we mentioned this briefly in the revised manuscript.

6.b) *Though clearly there is a limit where in the four-Fermi theory $S_e + S_1 + S_2$ the double fermion internal loop may be approximated by a single composite effective $N(S)$ or $D(d)$ line, it is not clear what this regime is. This should be expressly stated.*

Response: We agree with the referee that the presentation of Eq. (2) might have been confusing. The symbols $\mathbf{S}_q = \int_k c_{k+q}^\dagger s\tau_z c_k$ and $\mathbf{D}_q = \int_k c_{k+q}^\dagger s i s_y \tau_y c_{-k}^\dagger$ where just introduced to write the different terms of the four-fermion interaction in more compact form and not to introduce some approximate description in

terms of effective composite fields. We have revised the manuscript to make this clear to the reader.

6.c) It is unclear what is meant by "anomalous contributions".

Response: By "anomalous contributions", we refer to self-energy diagrams with two incoming or two outgoing fermionic lines (as opposed to one out-going and one incoming line); these diagrams are known to be associated with superconductivity and, e.g., appear very prominently in Eliashberg theory. We agree, however, with the referee that one should explain the meaning of the term, which is done in the revised version of the manuscript. In the previous version of the manuscript, we also used the term "anomalous interaction" once when referring to Eq. (2b); since this was not necessary, we changed that in the revised manuscript too.

6.d) Fig2d the term "self-consistent solution" is unexplained.

Response: In the revised manuscript, we have clarified the self-consistent solution to mean the full solution taking into account the higher order corrections to the self-energy. Thank you for pointing out that this was previously not clearly explained.

6.e.) The paragraph "Before proceeding" appears to be a computational detailed discussion best deferred to supp.mat. This is clearly illustrated by the use of the identical sentence "The anomalous contribution ... channel" in both the preceding and subsequent paragraph.

Response: We agree with the referee that this paragraph should be removed from the main text, which we have done in the revised version of the manuscript.

6.f) Though the analysis of the DOS should be there for the experimental reason highlighted by the authors, the section only gives mathematical explanations, but no physical ones for this effect.

7. In that sense the minimal mean field theory model is much more instructive.

Response: We agree with the referee that it would be helpful to add a more qualitative reason for why the resulting density of states is different from that of the usual BCS mean-field theory. To this end, we first note that, on a high level, the key difference to BCS mean-field theory, where an anomalous (i.e., particle-number non-conserving cf. 6c above) term of the form $\Delta c^\dagger c^\dagger + \text{H.c.}$ is added to the Hamiltonian, is that a term of the form $\phi_0 c^\dagger c^\dagger c^\dagger c + \text{H.c.}$ appears in phase (B). Being quadratic in BCS, Δ just leads to an avoided crossing between particle and hole bands and a hard gap opens, given by $|\Delta|$; diagrammatically one can think of Δ as being the (within mean-field theory exact) anomalous self energy. In our case of phase (B), the term is quartic, i.e., constitutes an interaction. This has crucial consequences which can be understood

as follows. First, in the diagrammatic approach, it means that the resulting self-energy is not just ϕ_0 , but (to leading order) a one-loop diagram involving a fermionic and bosonic propagator. This induces significant momentum and frequency dependencies in the anomalous self-energy although the underlying ϕ_0 was just a constant. It is these dependencies that lead to the different behavior of the density of states. Second, this can be made more concrete and intuitive in the mean-field approach (in line with what the referee writes): the extra factor of $c^\dagger c$ in $\phi_0 c^\dagger c^\dagger c^\dagger c + \text{H.c.} = c^\dagger c (\phi_0 c^\dagger c^\dagger + \text{H.c.})$ compared to BCS ultimately leads to two coupled (instead of a single) gap equations where [cf. Eq. (8b) in the revised manuscript] the effective mean-field superconducting order parameter $\tilde{\Delta}_{\mathbf{k}}$ is proportional to the renormalized dispersion $\tilde{\epsilon}_{\mathbf{k}}$ in the perturbative regime. This is why $\tilde{\Delta}_{\mathbf{k}}$ vanishes linearly with $\tilde{\epsilon}_{\mathbf{k}}$ at the Fermi surface ($\tilde{\epsilon}_{\mathbf{k}} = 0$) and as such only corresponds to partial rather than complete suppression of the spectral weight.

Since the physical origin of the differences in the DOS of phase (B) and BCS theory is indeed an important point in our work, we have added comments on it both in the diagrammatic section as well as in the mean-field theory.

I would urge the authors to revise and rewrite with the broader Nat.Comm. audience in mind, and resubmit.

Response: We have done our best to achieve this in the revised manuscript. We thank the referee for their comments, which were really helpful for making the manuscript more accessible to non-experts. With our modifications, we are confident that the manuscript is now suitable for the broader readership of Nature Communications.

Point-by-point response to referee 3:

This MS presents a comprehensive study of vestigial order in a 2D system with attraction in valley FM/inter-valley AMF channel and valley-singlet spin-triplet superconducting channel. Neither AFM nor B-type spin-triplet superconductivity are possible in 2D at a finite T , however composite (vestigial) orders are potentially possible as they do not break $O(3)$ continuous symmetry. Previous works on vestigial orders focused primary on the superconducting channel and argued that such order can be viewed as $4e$ superconductor. The authors analyzed in great detail different composite order - the one with the charge $2e$. They demonstrated that such an order is a spin-singlet 4-fermion bound state between a spin-triplet particle-particle pair and a $S=1$ soft particle-hole excitation associated with valley FM (3 electrons and one hole or 3 holes and one electron). They argued that the charge $2e$ bound state possesses quasi-long-range order at a finite T in 2D and displays a Meissner effect, like a conventional spin-singlet superconductor, but the suppression of the density of states at small frequencies is more complex than in a conventional case. The authors argue that their results are relevant to experiments on twisted bilayer graphene.

I truly enjoyed reading this paper. It is of high-quality, and the authors presented enough details of their calculations, so a reader can follow their analysis. My recommendation is to accept this MS to Nature Communications.

Response: We thank the referee for their time and are happy to hear that they recommend publication of our work in Nature Communications. We next address the referee's suggestions.

Two suggestions:

1. On the spin-triplet/valley singlet order with a constant gap. The authors probably meant the analog of a B-type phase for ^3He , not the A-type phase. I would suggest to add an explanatory phrase on this.

Response: Yes, indeed, the triplet state with order parameter $\mathbf{d} \cdot s_i \tau_y$ is more similar to the B phase rather than the A phase of superfluid Helium-3 in that it is fully gapped and does not exhibit nodal points. We note that our labeling of phases in Fig. 1 as (A), (B), and (C), has no relation to the nomenclature of Helium-3. We agree with the referee that it is a good idea to clarify this for the reader; we have added a comment in the section "Zero-temperature phases" in the revised manuscript.

2. About the one-loop expressions for the self-energy and the discussion after Eq. (3). I suggest to state explicitly that the peak in the spectral function is pushed away from $\omega = 0$ only if the coupling λ exceeds a certain limit.

Response: We thank the referee for this suggestion which we have taken into account in the revised version

of the manuscript: we mention this property of the spectral function when discussing Fig. 3(a,b) in the revised version of the manuscript.

List of changes made

1. We removed the reference to the phase diagram in Fig. 1 in the introduction and rearranged the order of (a) and (b,c) in Fig. 1. In this way, all symbols are well defined when we first refer to the phase diagram in Fig. 1 (in the results section).
2. In accordance with Nature Communications' formatting, we added dedicated titles (indicated in bold) to all figures.
3. We have added an explicit reference to the spin fermion model in the beginning of the section "Model".
4. We split the section "Possible vestigial phases" into two, "Zero-temperature phases." and "Vestigial phases at finite T " and revised the text in the following way:
 - (a) We now mention explicitly and prominently which part refers to $T = 0$ and that the remainder of the paper refers to $T > 0$.
 - (b) We refer earlier to Table I, to make sure that the symmetries are well defined when discussing Fig. 1(c).
 - (c) We state explicitly that the labeling of phases we use in Fig. 1 is not related to the nomenclature of ^3He .
 - (d) We explain the meaning of the key couplings b_2 and c_1 and how this related to the different phases.
5. In the paragraph above Eq. (1), we mention more explicitly that our (first) large- N approach does not involve taking infinitely many fermion flavors, but rather promoting the three-component bosons d and N to have infinitely many components.
6. We introduced the more intuitive notation of S_B and $S_{B'}$ to refer to the theory for phase (B) before and after integrating out the bosons.
7. We also state explicitly in the revised manuscript that the terms in Eq. (2) are four-fermion interactions.
8. We further explicitly mention that $M_q \rightarrow \chi_d^{-1} \chi_N^{-1}$ in the limit $\phi_0 \rightarrow 0$ below Eq. (2).

9. At the beginning of the section “Electronic self energy”, we explain what we mean by normal and anomalous self-energy contributions. Similarly, we changed “anomalous interaction” to “particle-number-non-conserving interaction” at the beginning of “Minimal mean-field theory”.
 10. We added a comment in the caption of Fig. 2 and in the section “Density of states” that explains what we mean by “self-consistent solution”.
 11. The paragraph in “Density of states” that, as correctly pointed out by referee 2 contained redundant and besides only technical information, was removed from the main text.
 12. In the section “Density of states”, we added a comment on the pinning of the peak in the spectral function at $\omega = 0$.
 13. We have added an intuitive discussion for why the behavior is different from the usual BCS theory in the penultimate paragraph of the section “Density of states” (on the level of diagrammatics) and in the section “Minimal mean-field theory” (on the level of mean field).
 14. We have revised the notation in the section “Minimal mean-field theory” to highlight better the connections with and difference to the diagrammatic/field-theoretical approach.
-

REVIEWERS' COMMENTS

Reviewer #1 (Remarks to the Author):

The paper by P.P. Poduval and M.S. Scheurer is a high-quality, thorough study with detailed calculations. Regrettably, I must re-iterate my earlier subjective opinion that the findings lack sufficient significance for publication in Nature Communications.

To recap, this manuscript reports two principal findings (potentially suitable for two separate publications):

1. It predicts a phase characterized by a non-zero product $N \cdot d$, comprising the magnetic order parameter N and the spin-triplet superconducting parameter d in a twisted graphene bilayer system.
2. It investigates the density of states (DoS) within this phase, showing a partial or complete suppression in line with the parameter range. This suppression qualitatively parallels the DoS observed in experimental measurements. The authors advocate for this finding as both a validation of the predicted phase and a plausible interpretation of experimental data.

Regarding the first finding, I consider the prediction somewhat trivial. The manuscript appears to postulate rather than derive this phase. The authors emphasize in their rebuttal that this postulation is supported by the sign of the mean-field results for the coefficient c_1 in the Ginzburg-Landau expansion, and they provide a discussion about the phase's compatibility with the Mermin-Wagner theorem. Nevertheless, I remain convinced of the insufficient significance of these results for Nature Communications.

With respect to the second finding, the calculations presented are meticulous and substantive. However, the association with experimental results seems tenuous and speculative. As acknowledged by the authors, partial suppression of the DoS is a common characteristic in interacting metals. Such DoS behaviors can arise from various interactions and mechanisms in metals and nodal systems.

In objection to that, the authors say in their response that the experimentally observed crossover from V-shaped to U-shaped DoS occurs in a superconducting state, which would typically exhibit either V- or U-shaped DoS depending on the superconductivity type, but not a crossover between them. However, this objection does not fully persuade me. The calculations in the paper are based on a phenomenological model where free electrons interact with order-parameter fluctuations, akin to the spin-fermion model. Perturbative calculations of the DoS in this model closely resemble the process of estimating the DoS in metals with Coulomb interactions. This similarity suggests that a variety of interactions and order parameters could result in a dip in the DoS. The transition from V-shaped to U-shaped DoS with increasing coupling, particularly in a narrow-band system, is not an unexpected finding. Even if the proposed mechanism were responsible for the experimental behavior observed, the finding would not be surprising given the initial model. Furthermore, there are many other potential explanations for the observed behavior, such as nodal superconductivity combined with fluctuations in another order parameter or the effects of impurities. In summary, while I appreciate the quality of the calculations, I find the manuscript's connection to experimental results too tenuous. This perspective is, of course, subjective.

Consequently, I believe the manuscript is not a fit for Nature Communications. However, I do see its potential as excellent material for Physical Review, possibly split into two separate publications.

Reviewer #2 (Remarks to the Author):

Having read the resubmitted version of "Vestigial singlet pairing in a fluctuating magnetic triplet superconductor: Implications for graphene superlattices" I happily reiterate my first impression: "The key result of this article is the unveiling of intertwined of vestigial order between triplet superconductivity and antiferromagnetic order, which is highly likely to be relevant for as yet unexplained experimental observations in bilayer graphene.

This work (and I would venture follow-ups thereof by the authors or others) are likely to be of prominent significance in this field. As the authors describe, bilayer graphene systems show a very rich variety of simple and complex phases and many of the latter could be of the intertwined type where this type of analysis applies."

I was very glad to see that the authors took my and the other reviewers comments and suggestions seriously. They significantly enhanced the readability of the article — for almost all of it. The sections Theory for Phase (B); Electronic self-energy; Density of States are very dense on detailed calculational information that is more appropriate for a specialized journal. (Also structurally the sections "the electronic self-energy"; "density of states" and "minimal mean field theory" are hierarchically below "Theory for Phase B" rather than at the same level). I would strongly advise to put the digestible self-consistent mean field theory section before the electronic self-energy; density of states , and condense the latter two (with perhaps some part relegated to supp.mat.). With that admonition, I would recommend publication in Nat.Comm.

Minor questions.

(1) I recommend renaming χ^0_{μ} as $\chi^{\{\text{bare}\}}_{\mu}$ to avoid confusion with the static susceptibility $\chi(q=0)$ which is often denoted as χ^0 .

(2) There is a typo in the self-consistent mean-field theory section: superconctivity \rightarrow superconductivity.

Reviewer #3 (Remarks to the Author):

I read the new MS and the authors reply to my questions and to questions/comments by other referees.

In my view, the authors gave adequate reply to the referees.

I do not agree with the first referee that the paper is not suitable for Nat. Communications. I found on the contrary that it presents enough new physics to fully warrant the acceptance.

The second referee suggested rewriting parts of the text for better presentation, and the authors did this.

The answers to my questions are satisfactory.

I recommend accepting this MS for publication. All three referees agree that this is high-quality work.

**Resubmission of NCOMMS-23-32697A:
“Vestigial singlet pairing in a fluctuating magnetic triplet superconductor:
Implications for graphene superlattices”**

Prathyush P. Poduval and Mathias S. Scheurer

(Dated: January 24, 2024)

Dear Dr. Paul Wiecki,

Thank you very much for your message on January 10 and for accepting our paper, in principle, for publication.

In this re-submission, we have addressed all editorial requests (see checklist, revised manuscript files, and Supplementary Information). Referee # 3 has no remaining open questions and suggests publication of the manuscript in its current form. Referee # 1 does not contain any constructive criticism that can be addressed and just repeats their previous general comments (which he/she admits are subjective). Only referee # 2, who recommends publication of our manuscript, has comments (two trivial typos and a suggestion of reordering two sections) that can be addressed. We have fixed the typos and reordered the sections as suggested, which indeed makes the presentation more accessible.

As such, we believe that the revised version of the manuscript can be published in Nature Communications in its current form.

Yours sincerely,

Prathyush P. Poduval and Mathias S. Scheurer

Point-by-point response to referee 1:

The paper by P.P. Poduval and M.S. Scheurer is a high-quality, thorough study with detailed calculations. Regrettably, I must re-iterate my earlier subjective opinion that the findings lack sufficient significance for publication in Nature Communications.

To recap, this manuscript reports two principal findings (potentially suitable for two separate publications):

1. It predicts a phase characterized by a non-zero product $N \cdot d$, comprising the magnetic order parameter N and the spin-triplet superconducting parameter d in a twisted graphene bilayer system.

2. It investigates the density of states (DoS) within this phase, showing a partial or complete suppression in line with the parameter range. This suppression qualitatively parallels the DoS observed in experimental measurements. The authors advocate for this finding as both a validation of the predicted phase and a plausible interpretation of experimental data.

Regarding the first finding, I consider the prediction somewhat trivial. The manuscript appears to postulate rather than derive this phase. The authors emphasize in their rebuttal that this postulation is supported by the sign of the mean-field results for the coefficient c_1 in the Ginzburg-Landau expansion, and they provide a discussion about the phase's compatibility with the Mermin-Wagner theorem. Nevertheless, I remain convinced of the insufficient significance of these results for Nature Communications.

With respect to the second finding, the calculations presented are meticulous and substantive. However, the association with experimental results seems tenuous and speculative. As acknowledged by the authors, partial suppression of the DoS is a common characteristic in interacting metals. Such DoS behaviors can arise from various interactions and mechanisms in metals and nodal systems.

In objection to that, the authors say in their response that the experimentally observed crossover from V-shaped to U-shaped DoS occurs in a superconducting state, which would typically exhibit either V- or U-shaped DoS depending on the superconductivity type, but not a crossover between them. However, this objection does not fully persuade me. The calculations in the paper are based on a phenomenological model where free electrons interact with order-parameter fluctuations, akin to the spin-fermion model. Perturbative calculations of the DoS in this model closely resemble the process of estimating the DoS in metals with Coulomb interactions. This similarity suggests that a variety of interactions and order parameters could result in a dip in the DoS. The transition from V-shaped to U-shaped DoS with increasing coupling, particularly in a narrow-band system, is not an unexpected finding. Even if the proposed mechanism were responsible for the experimental behavior observed, the finding would not be surprising given the initial

model. Furthermore, there are many other potential explanations for the observed behavior, such as nodal superconductivity combined with fluctuations in another order parameter or the effects of impurities. In summary, while I appreciate the quality of the calculations, I find the manuscript's connection to experimental results too tenuous. This perspective is, of course, subjective.

Consequently, I believe the manuscript is not a fit for Nature Communications. However, I do see its potential as excellent material for Physical Review, possibly split into two separate publications.

Response: We are happy to hear that the referee still believes that our work is “a high-quality, thorough study with detailed calculations”. In line with the other two referees, who recommend publication in Nature Communications, we however strongly disagree with referee # 1 that our manuscript is not a good fit for Nature Communications. As referee # 1 him/herself admits, this “perspective is, of course, subjective”. As such and since the referee basically repeats their general points from the previous round of review, we can only refer back to our first response where we argued why our results are very much interesting enough for publication in Nature Communications and not “somewhat trivial”.

Point-by-point response to referee 2:

Having read the resubmitted version of “Vestigial singlet pairing in a fluctuating magnetic triplet superconductor: Implications for graphene superlattices” I happily reiterate my first impression: “The key result of this article is the unveiling of intertwined of vestigial order between triplet superconductivity and anti-ferromagnetic order; which is highly likely to be relevant for as yet unexplained experimental observations in bilayer graphene. This work (and I would venture follow-ups thereof by the authors or others) are likely to be of prominent significance in this field. As the authors describe, bilayer graphene systems show a very rich variety of simple and complex phases and many of the latter could be of the intertwined type where this type of analysis applies.”

I was very glad to see that the authors took my and the other reviewers comments and suggestions seriously. They significantly enhanced the readability of the article — for almost all of it. The sections Theory for Phase (B); Electronic self-energy; Density of States are very dense on detailed calculational information that is more appropriate for a specialized journal. (Also structurally the sections “the electronic self-energy”; “density of states” and “minimal mean field theory” are hierarchically below “Theory for Phase B” rather than at the same level). I would strongly advise to put the digestible self-consistent mean field theory section before the electronic self-energy; density of states , and condense the latter two (with perhaps some part relegated to supp.mat.). With that admonition, I would recommend publication in Nat.Comm.

Response: We thank the referee for their comments and positive evaluation of the work. As suggested by the referee, we have changed the order of the sections, such that the self-consistent mean-field calculation is discussed first.

Minor questions.

(1) I recommend renaming χ_{μ}^0 as χ_{μ}^{bare} to avoid confusion with the static susceptibility $\chi(q = 0)$ which is often denoted as χ^0 .

(2) There is a typo in the self-consistent mean-field theory section: superconctivity \rightarrow superconductivity.

Response: Thank you for reading our manuscript carefully. We have fixed the typo (2) and (1) agree with the referee that χ_{μ}^0 can be confused with the bare susceptibility; therefore, we have replaced it with the variable $\bar{\chi}_{\mu}$.

Point-by-point response to referee 3:

I read the new MS and the authors reply to my questions and to questions/comments by other referees.

In my view, the authors gave adequate reply to the referees. I do not agree with the first referee that the paper is not suitable for Nat. Communications. I found on the contrary that it presents enough new physics to fully warrant the acceptance.

The second referee suggested rewriting parts of the text for better presentation, and the authors did this.

The answers to my questions are satisfactory.

I recommend accepting this MS for publication. All three referees agree that this is high-quality work.

Response: We thank the reviewer for their positive evaluation of our work and their comments.